# Nanotechnology Incorporation into Road Pavement Design Based on Scientific Principles of Materials Chemistry and Engineering Physics Using New-Age (Nano) Modified Emulsion (NME) Stabilisation/Enhancement of Granular Materials

**Gerrit J. Jordaan** [1,2,*] and **Wynand J. vdM Steyn** [3]

1   Department of Civil Engineering, University of Pretoria, Pretoria 0002, South Africa
2   Jordaan Professional Services (Pty) Ltd., Pretoria 0062, South Africa
3   School of Engineering and Department of Civil Engineering, University of Pretoria, Pretoria 0002, South Africa; Wynand.steyn@up.ac.za
*   Correspondence: jordaangj@tshepega.co.za; Tel.: +27-(0)-824164945

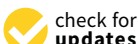



**Featured Application: Most work throughout the world on the use of nanotechnology solutions in pavement engineering has been concentrated on the improvement of asphaltic materials. The ability presented by the use of proven applicable nanotechnology solutions to enable naturally available materials to be enhanced, protected and utilized in all the structural layers of pavements has received little attention. Over the last 7 years, design procedures have been developed, laboratory evaluated, implemented in the field and evaluated using full-scale Accelerated Pavement Tests (APT). Results have exceeded all expectations with the design traffic loadings being achieved and exceeded by some considerable margins. This article presents the materials design procedure developed, which is based on universally applicable scientific principles, incorporating materials chemistry and recommends specifying end-product requirements in terms of fundamental engineering physics.**

**Abstract:** The use of naturally available materials not conforming to traditional specifications or standards in the base and sub-base layers of road pavement structures and stabilised with New-age (Nano) Modified Emulsions (NME) have been tested, implemented and successfully verified through Accelerated Pavement Testing (APT) in South Africa. This was made possible through the development and use of a materials design procedure addressing fundamental principles and based on scientific concepts which are universally applicable. The understanding and incorporation of the chemical interactions between the mineralogy of the materials and an NME stabilising agent (compatibility between the chemistry of the reactive agents and material mineralogy) into the design approach is key to achieving the required engineering properties. The evaluation of the stabilised materials is performed using tests indicative of the basic engineering properties (physics) of compressive strengths, tensile strengths and durability. This article describes the basic materials design approach that was developed to ensure that organofunctional nano-silane modified emulsions can successfully be used for pavement layer construction utilising naturally available materials at a low risk. The enablement of the use of naturally available materials in all pavement layers can have a considerable impact on the unit cost and lifecycle costs of road transportation infrastructure.

**Keywords:** road pavement design; design based on materials science; material mineralogy; new-age (nano) modified emulsions (NME); naturally available materials; material stabilisation; basic engineering requirements; unconfined compressive strengths (UCS); indirect tensile strengths (ITS); retained compressive strengths (RCT); retained tensile strengths (RTS)

## 1. Introduction

Organofunctional nano-silane technologies have been used in Europe for the protection of stone buildings for more than 150 years [1,2]. The application of various nano-silane products has been used to protect buildings against the climatic effects of moisture, providing a hydrophobic protective layer and preventing further decay due to chemical weathering. The initial work performed by scientists to develop protective products relied on trial-and-error testing and resulted in contradictorily results, as reported by various scientists. Eventually, it was concluded that the successful application of a silane-based protective product depends to a large extent on the compatibility of the nano-silane product with the "type of stone" as well as the "condition of the stone" to be treated [3]. The technology developed in the built environment, as well as the "lessons learnt", can find direct application and should form the basis for the successful introduction and treatment of available materials in road pavement engineering [4].

The potential impact of the use of nanotechnologies in the field of pavement engineering was identified more than a decade ago [5]. However, in practice, the introduction of applicable and proven nanotechnologies has been slow to receive acceptance. This scepticism in the pavement engineering fraternity has its origin in a constant flow of "wonder" products (commonly referred to as snake-oils) that have been introduced into the market by suppliers for the improvement of naturally available materials. These products have invariably been found wanting in practice, not meeting the anticipated results and leading to a general scepticism that is detrimental to the introduction of new technologies into the field of pavement material stabilisation. Both the suppliers of new products/technologies as well as the road pavement engineering fraternity are to blame for this situation. Pavement engineers mostly rely on archaic, empirically derived tests (some more than a hundred years old), not indicative of the scientific characteristics of materials [6]. These tests are found wanting in the assessment of new products and the evaluation of their true potential and limitations in terms of sound engineering requirements. In the same vein, suppliers of new products invariably lack a scientific basis and the data to support their claims, often using marketing agents with little engineering knowledge. This situation is very similar to what scientists in the built environment experienced more than 150 years ago with the development of stone protectors on a trial-and-error basis.

Over the last few decades, considerable resources in pavement engineering have been allocated to the development of sound Mechanistic-Empirical design methods to improve the analysis of pavement structures. Transfer functions based on the stresses, strains and deformation characteristics of various materials have been developed to improve the scientific basis of pavement design, identifying failure mechanisms and developing failure theories for universal use and analysis. The same (with some notable exceptions such as in the field of asphaltic materials) cannot be said about the testing, evaluation and characterisation of the materials used in the construction of road pavement structures. Most of the granular material characterisation tests lack a scientific basis with little relation to potential bearing capacity and the theoretical analysis of pavement structures. This lack of sound engineering concepts for the characterisation of naturally available materials and the specification of engineering requirements in terms of basic physics requires urgent attention. The observation is not new. The inconsistencies between a "petrological, genetically-based classification" [7] and the engineering classification of materials for road construction had already been recognised by the British Standards Institution in 1954 [8] and also (inter alia) by the South African Bureau of Standards in 1976 [9], with little impact on traditional engineering practices.

The "lessons learnt" from the built environment should form the basis of any design method aimed at the successful introduction of proven and available new-age nanotechnology solutions in the field of pavement engineering. Without an understanding of the basic science in terms of the mineralogy of the materials and the chemical interaction with organofunctional nano-silane based products and stabilising agents, the introduction and acceptance of the benefits to be achieved through the introduction of these technologies

could be lost to the industry. This is mainly due to the risks and potential failures associated with the use of products that are not materially compatible. A scientifically based materials design method for the appropriate use of applicable nano-silane products will limit these risks and overcome the limitations of an empirically derived materials classification system.

A scientifically based design method is universally applicable and does not require verification under different conditions. Such a method must be based on scientifically determined input values. A nano-silane modification of a stabilising agent of road building materials is reactive in nature, the influence of which depends on the chemical interaction with the minerals within the materials. Hence, to limit risks and address engineering requirements, any design process must be based on the scientific identification of the primary and secondary minerals in the material [10] and an understanding of the basic chemistry involved in the use of nano-silane reactive agents with the minerals [11]. In order to address basic fears associated with new technologies, fundamental principles and evaluation criteria need to be identified and adequately addressed [12]. Evaluation criteria should be based on fundamental engineering properties used to assess the adequacy of the future behaviour in a pavement structure in terms of compressive strengths, tensile strengths and durability (i.e., resistance to in situ deterioration as a function of climate and loading conditions).

Over and above the technical merits, the successful implementation of new technologies is also a function of practical considerations, such as costs and ease of use and application (e.g., stability in often harsh conditions) and constructability [4]. All practical and functional factors need to be identified and accounted for [12] to ensure that the substantial benefits that can be realised through the introduction of applicable nanotechnologies in the field of pavement engineering can be successfully introduced and find general acceptance for the benefit of society as a whole [4,6,13].

In the development of a scientifically based materials design approach, considerable effort has been put into the understanding of the basic principles and the "lessons learnt" from the built environment, which have been dealt with in detail in associated articles [4,10]. The objective of this article is to combine all the factors mentioned in a scientifically based materials design method, which can be applied universally, that will:

- Enable naturally available materials to be fully utilized at a fraction of the cost of imported, newly crushed stone;
- Meet the engineering requirements in terms of fundamental physics, i.e., stresses, strains and durability;
- Accommodate ease of use to be applied within a low-risk environment without compromising quality.

This article outlines the major components of a materials design process developed to ensure that new-age nanotechnology solutions are successfully introduced into the field of pavement engineering, meeting the objectives.

## 2. Background—Primary Benefits of the Introduction of New-Age Nanotechnology Solutions (Nano-Silanes) in Pavement Materials Design in Pavement Layers

Traditionally, materials for use in road construction are classified based on empirically derived archaic tests, aimed at the minimisation of the risk of failure during the design period, assuming future adequate preventative maintenance [6]. These classification systems mainly aim to ensure that the presence of secondary minerals (the result of the chemical decomposition of the primary minerals), which could be harmful to the future performance of materials in pavement structures, is kept at a minimum. The resulting effect is that these criteria will mostly only be met by freshly crushed stone for use in the upper pavement layers (in pavement structures with thin asphaltic surfacings or chip seal surfacings) directly subjected to the high stresses/strains imposed by the traffic loading. The blasting and crushing of freshly crushed stone come at a considerable cost with an associated environmental impact and is a scarce commodity in many parts of the world. The result is that the unit costs of roads are considerable, making it highly improbable for

the developing world to substantially increase their surfaced road network in support of economic development in the foreseeable future.

The only alternative available option currently used in the development of a surfaced road network in many sub-Saharan countries is the stabilisation of quality naturally available materials using traditional stabilising agents. This practice usually involves stabilisation through the use of cement and/or lime with associated construction management (curing), material behaviour challenges (e.g., cracking) and material compatibility problems. These traditional stabilising agents (also containing nano-scale particles) are also reactive agents, often resulting in severe cracking and premature distress that cannot be explained using the results obtained from traditional test methods. In reality, the premature distress can often be associated with the mineralogy of the materials (e.g., the presence of mica [14]), which is not accounted for using traditional design material testing.

Most work throughout the world on the use of nanotechnology solutions in pavement engineering has been concentrated on the improvement of binders for asphaltic materials [15,16]. This trend is no surprise due to the popularity of the concept of full-depth asphalt pavements, the vulnerability of asphalt binder characteristics to the effects of aging (and associated problems [4]) and the considerable investments needed in the construction of full-depth asphalt pavements. In order to reduce the costs of full-depth asphalt roads, considerable investments in research have been made since the 1970s for alternative technologies and the use of freshly crushed stone as a high-quality alternative in combination with relatively thin asphaltic layers or even chip seal surfacings. This evolution in road pavement design has led to considerable cost-saving for high-end roads, as reported in [17]. In later years, the misuse of this technology has led to unaffordable high unit costs, with high-quality crushed stone becoming the norm for all categories of roads. The unit costs for the provision of road infrastructure, even using crushed stone as an alternative to full-depth asphalt roads, has become a major obstacle to development and to the accessibility to markets [18]. However, material sciences have made some significant advances over the last few decades that should be the key to the next evolution in the use of available granular materials in pavement design.

Organofunctional nano-silane technologies have the ability to neutralise the presence of potentially harmful secondary minerals in materials. Through the application of a material-compatible reactive nano-silane, a chemical reaction [11] with each material particle can be activated to change the surface characteristics to become hydrophobic (water repellent) and prevent the possible negative impact of secondary minerals on the future performance of a road structure [4,12]. In addition, future weathering through chemical decomposition is prevented, or at least minimised, due to the repelling of water or moisture within the pavement structure. The presence of water is a pre-requisite for weathering due to the chemical decomposition that occurs [7]. It follows that the durability of the materials within the pavement structure will be increased by minimising chemical decomposition. This can be achieved through the introduction of minimum criteria, limiting the effect of the influence of water submersion on test samples on the compressive and tensile strengths of materials. This aspect can also be key to the successful treatment of materials in regions of the world subjected to frost-freeze conditions.

The nano-silane in itself is not effective nor used as a stabilising agent. The nano-scale particles are too small to bridge the gaps between the granular particles of the road-building materials. Hence, it is used as a modification to a material-compatible stabilising agent (e.g., bitumen or equivalent polymer) in an emulsion form to "permanently" attach the stabilising agent (with a larger particle size) to the road-building materials to form a highly flexible durable layer or layers within a pavement structure [4].

The modification of an emulsion through the addition of nano-silane in effect introduces a second emulsifying agent with different characteristics into the mix, leading to a reduction in particle size (e.g., modification of bitumen emulsion) [19,20]. Smaller particle sizes within the modified emulsion have several advantages [6], including the improved stability of the modified emulsion and the ease of distribution of the emulsion within the

construction water, with related advantages during construction. Hence, the advantages of the introduction of material-compatible new-age nanotechnologies into the field of pavement engineering include:

- Cost factors: naturally available materials can be used at a low risk with an enhanced durability, reducing materials costs with savings in the procurement, transportation and construction costs associated with high quality crushed stone;
- Environmental factors: naturally available materials require less energy for production, i.e., the blasting, transportation, crushing and screening is normally not required. New-age (Nano) Modified Emulsions (NME) stabilisation eliminates the use of cement (the manufacturing of which is a considerable contributor to greenhouse gasses in the world, with an additional associated cost of around USD 70 to USD 80 per ton during manufacturing);
- Energy factors: the constructability of NME stabilisation is less complex, requiring considerably less compaction effort, i.e., energy (due to the introduction of a lubricant as part of the NME), compared to traditional high quality crushed stone layers and requires no heating when added as part of the construction water, resulting in considerable savings in energy.

Numerous different materials from all over southern Africa have been tested, determining the primary and secondary minerals through X-Ray Diffraction (XRD) scans [4] and successfully evaluated using tests indicative of basic engineering parameters. These include compressive strength, tensile strength and durability (assessing the hydrophobic effect achieved with the modified stabilising agent [12]) to ensure that the risk profile in terms of the category of road is met over the design period [18,21]. These materials vary from material with CBR > 45 per cent at 95 per cent mod AASHTO [22,23], to fine Kalahari Sands to highly weathered granites and laterites and include materials containing a high percentage of clays. The design process has been successfully applied in practice [24], evaluated using laboratory tests (e.g., [25]) and validated using Accelerated Pavement Tests (APT) [26,27].

## 3. Pavement Design Process Based on Scientific Principles

### 3.1. Elements Comprising the Design Approach

In order to incorporate the basic principles of science into pavement materials design, engineers need to understand and recognise the influence of the chemical interaction between materials and the influence thereof on the end product [11]. For this reason, some effort is made to include the basic concepts of primary minerals and the process of weathering due to chemical decomposition as a foundation for the understanding of the need for a scientifically based materials design approach in pavement engineering. The main phases of the design approach presented to facilitate the successful implementation of applicable and proven nanotechnology solutions (or for that matter any traditionally used or future new re-active agent) in road pavement materials engineering, are shown in Figure 1.

Basic inputs such as traffic loading, required as part of pavement structural design, will, of course, be part of the pavement design process and are not affected by the concepts discussed in this paper. Traffic loading is used to design a pavement structure with material layers meeting the engineering requirement to ensure that the predicted traffic loading will be carried, taking into consideration the risk profile of the category of road [21]. The main objective in the approach precented in Figure 1 is to ensure that the naturally available materials are utilised to their full potential using available, proven, safe and applicable material sciences in designing the most cost-effective road infrastructure at a minimum risk without compromising the fundamental engineering principles.

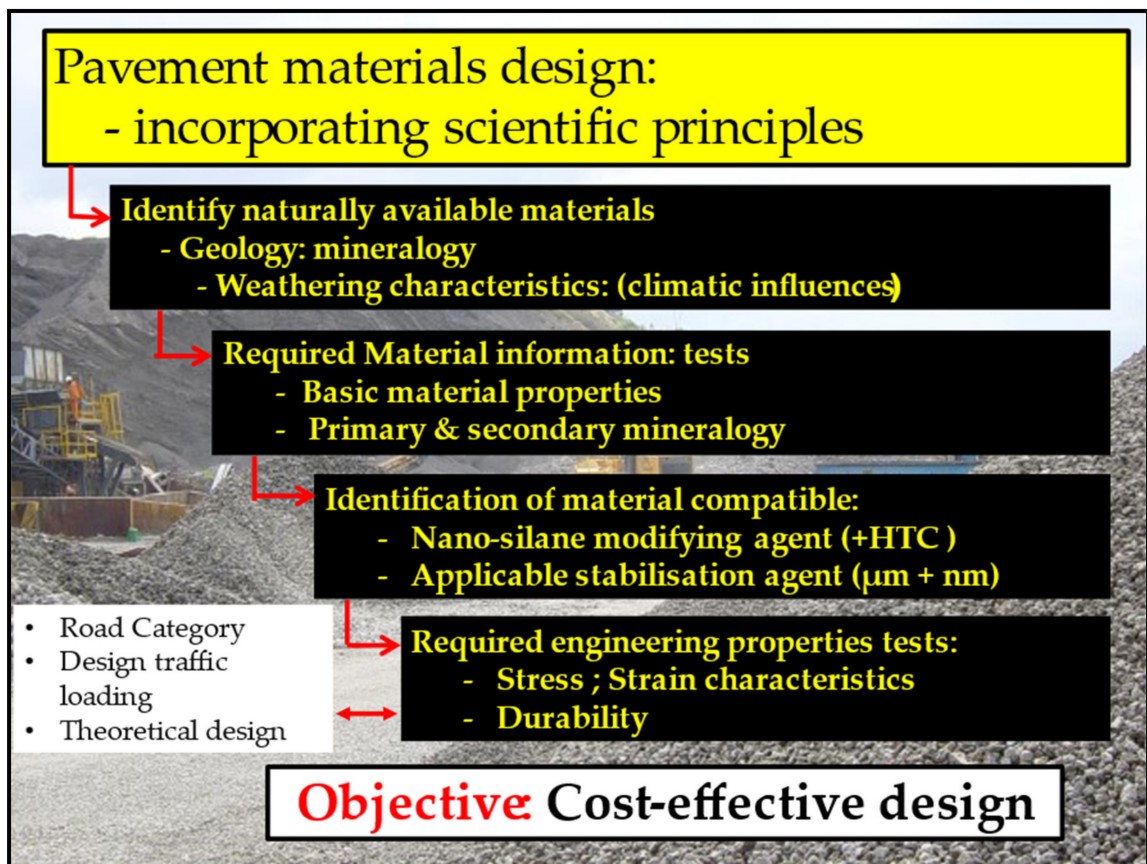

**Figure 1.** Recommended basic approach to road pavement materials design incorporating scientifically based concepts, developed for the selection of a material-compatible New-age (Nano) Modified Emulsion (NME) stabilising agent addressing the engineering requirements (physics) of a road pavement structure.

### 3.2. *Scientifically Based Material Properties of Naturally Available Materials*

3.2.1. Mineralogy of Naturally Available Materials

It is not the intention to replace the role of the geologist or to provide a comprehensive and detailed overview or description of all the various petrology and geology concepts as available in numerous sources. It is rather the intention to provide a simplistic approach with the emphasis on the basic concepts and understanding of the important role of mineralogy, in the successful implementation of proven and applicable nanotechnologies in the cost-effective use of naturally available materials in pavement engineering. The understanding of the roles of both the primary minerals ("type of stone") as well as the secondary minerals ("condition of the stone") are fundamental to the successful implementation of nanotechnologies through the incorporation of basic material science.

Engineers traditionally assume that materials classified on the basis of various empirically derived tests stay unchanged during the design period of a pavement structure. However, due to the presence of moisture and temperature fluctuations, the chemical weathering of materials is a natural, continuous process, even within a pavement structure. The material chemical decomposition is independent of the role in which it is used as long as it has access to water (moisture or vapour) and subject to temperature variations during seasonal changes. These influences are more pronounced in areas around the world characterised by considerable fluctuation in temperature, humidity and rainfall, as mostly experienced between the Tropic of Cancer (North) and Tropic of Capricorn (South). Some primary minerals have a higher resistance to chemical weathering, and a basic knowledge and an understanding of these characteristics are of importance in the identification of material-compatible nanotechnology solutions.

The naturally available materials precent between the two Tropics are usually the result of considerable chemical decomposition, representing a classification often more closely associated with soils than with the original geology. It follows that not only is the geology of importance but also the identification of soil-types. Geological and soils maps can be of great assistance to identify variations in the characteristics and the mineralogy of the materials that may be naturally available in any specific region [18]. The following factors all contributes to the mineralogy of the naturally available materials:

- Original rock formation and primary minerals;
- Climatic factors including rainfall, temperature and humidity;
- Changes in the slope of the area and vegetation that both contribute to the removal of the top, most weathered materials through the flow of water;
- Human interference through deforestation and other activities.

The variation in these factors has a considerable influence on the expected chemical weathering of the materials and changes in the expected fundamental mineral properties pavement materials engineers have to deal with, as summarized in Table 1 (summarised and based on [28]).

**Table 1.** Generalised material properties associated with different climatic regions of the world due to the effect of chemical weathering (based on [28]).

| Climatic Conditions | Cold Regions (Little Decomposition) | Warm Regions (Considerable Decomposition) |
| --- | --- | --- |
| **Property** | Materials: Conventional (Crushed rock base, river gravels, glacier outwash) | Materials: Pedogenic (laterites, calcretes, ferricretes, silcretes, etc.) |
| **Climate** | Temperate to cold | Arid, tropical, warm temperate |
| **Material Composition** | Natural or crushed | Varies from rock to sand to clay with considerable variation in each |
| **Material Chemical Reactivity** | Inert | Reactive |
| **Material Variability** | Homogeneous | Extremely variable |

### 3.2.2. Primary Minerals Defining the Type of Stone

Stone/gravel/soil/clay consists of numerous minerals that are influenced in different ways through the differences in climatic variations. More than 4000 minerals have been identified by scientists within the crust of the Earth, resulting in countless variations and permutations that could influence material behaviour [29]. However, the understanding of mineralogy can be considerably simplified through the identification of the minerals that are in abundance and form most of the crust of the Earth. It follows that these minerals will also be of particular importance in the identification of material-compatible nano-silane technologies to be used together with an appropriate stabilising agent. Only a few elements form the bulk of these minerals (Table 2 [30]). A generalised composition of Igneous rocks (solidification of lava materials) found in the crust of the earth is shown in Figure 2 (Compiled from various similar figures in public domain (e.g., [31]).

**Table 2.** Estimated percentages of the main elements found in the crust of the Earth that constitutes most of the materials (estimates based on [30])—some significant variances are precent around the globe.

|  | Estimated % by Weight | Estimated Atomic % | Estimated Volume % |
|---|---|---|---|
| **Oxygen (O)** | ~46.6–49.5 | ~62.3–66.2 | ~92–94 |
| **Silicon (Si)** | ~25.3–27.7 | ~19.4–21.2 | ~8–6 |
| **Aluminium (Al)** | ~7.5–8.1 | ~6.0–6.5 | |
| **Iron (Fe)** | ~4.2–5.1 | ~1.6–1.9 | |
| **Calcium (Ca)** | ~3.2–3.6 | ~1.7–1.9 | |
| **Sodium (Na)** | ~2.4–2.9 | ~2.0–2.4 | |
| **Potassium (K)** | ~2.3 - 2.6 | ~1.2–1.4 | |
| **Magnesium (Mg)** | ~1.9–2.1 | ~1.6–1.8 | |
| **All Others** | ~2.5 | ~0.5 | |
| **Total** | 100 | 100 | 100 |

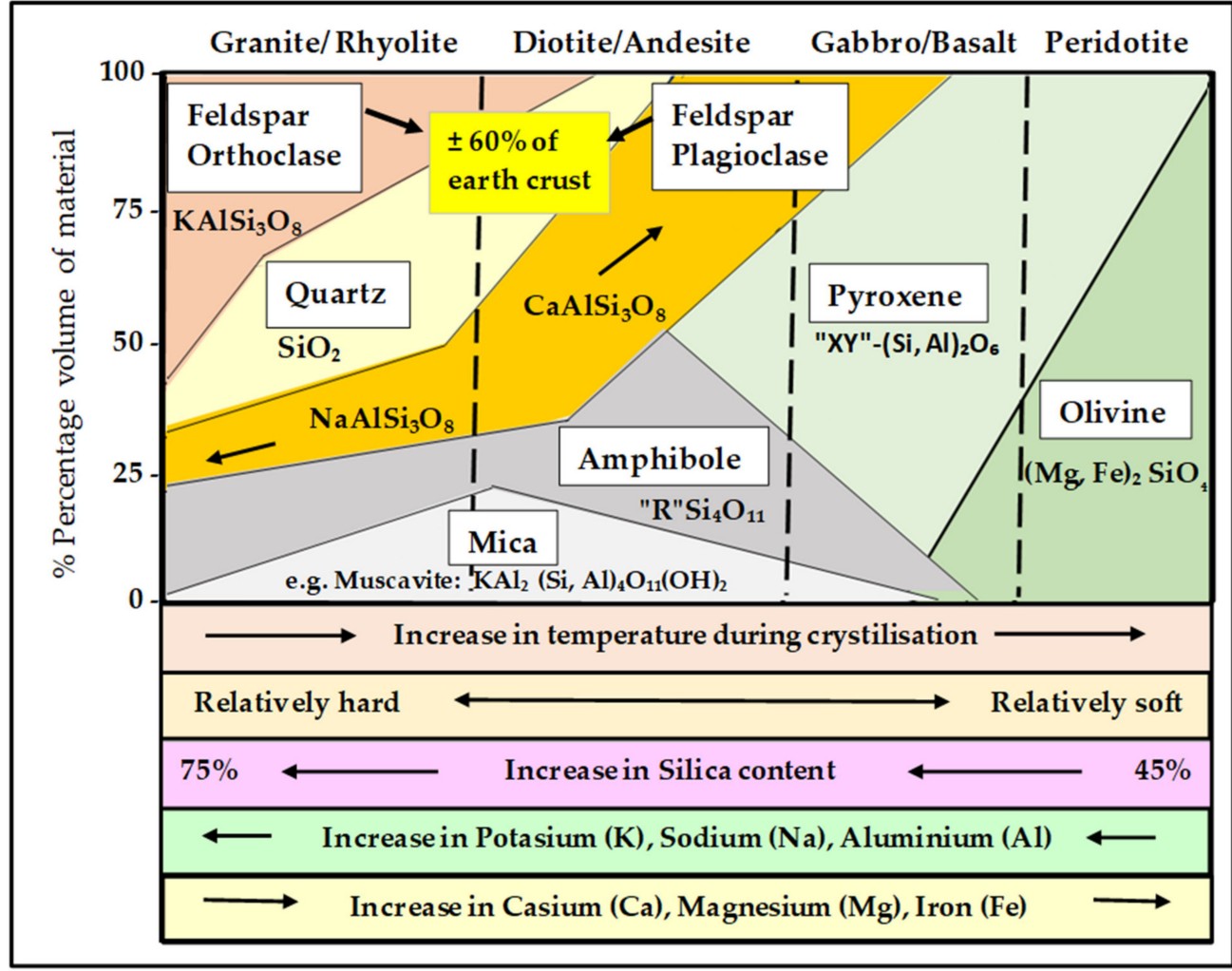

**Figure 2.** Generalised composition of Igneous rocks found in the crust of the Earth (based on and compiled from various similar figures in public domain, e.g., [31]).

Due to the abundance of the Silicon (Si) and Oxygen (O) elements (Table 2), more than 90 per cent of the materials found in the crust of the earth are defined as Silcretes. Hence, materials are generally defined as Silicate minerals and non-Silicate minerals (e.g., Carbonates, Oxides Sulphates and Halite). Since the Si–O bonds are some of the strongest in nature [11], it could easily be assumed that any modified stabilising agent compatible with Silicate materials should be effective and able to form strong bonds with most of the naturally available materials. However, such a generalised deduction is erroneous and could have severe detrimental consequences. Figure 3 give an indication of the Silicon versus Carbon-dioxide content of some of the more well-known commonly found naturally available materials. Figure 4 [32], for example, gives the lithological properties of the surface geology of the African continent, clearly showing that large areas of the surface geology of the continent, especially around southern Africa, the Horn of Africa and norther Africa contain Carbonate materials not suitable for NME stabilisation using a conventional nano-silane modification. The surface material distribution is not unique to the African continent, as shown by the soil distribution map of Africa in a global context in Figure 5 [32].

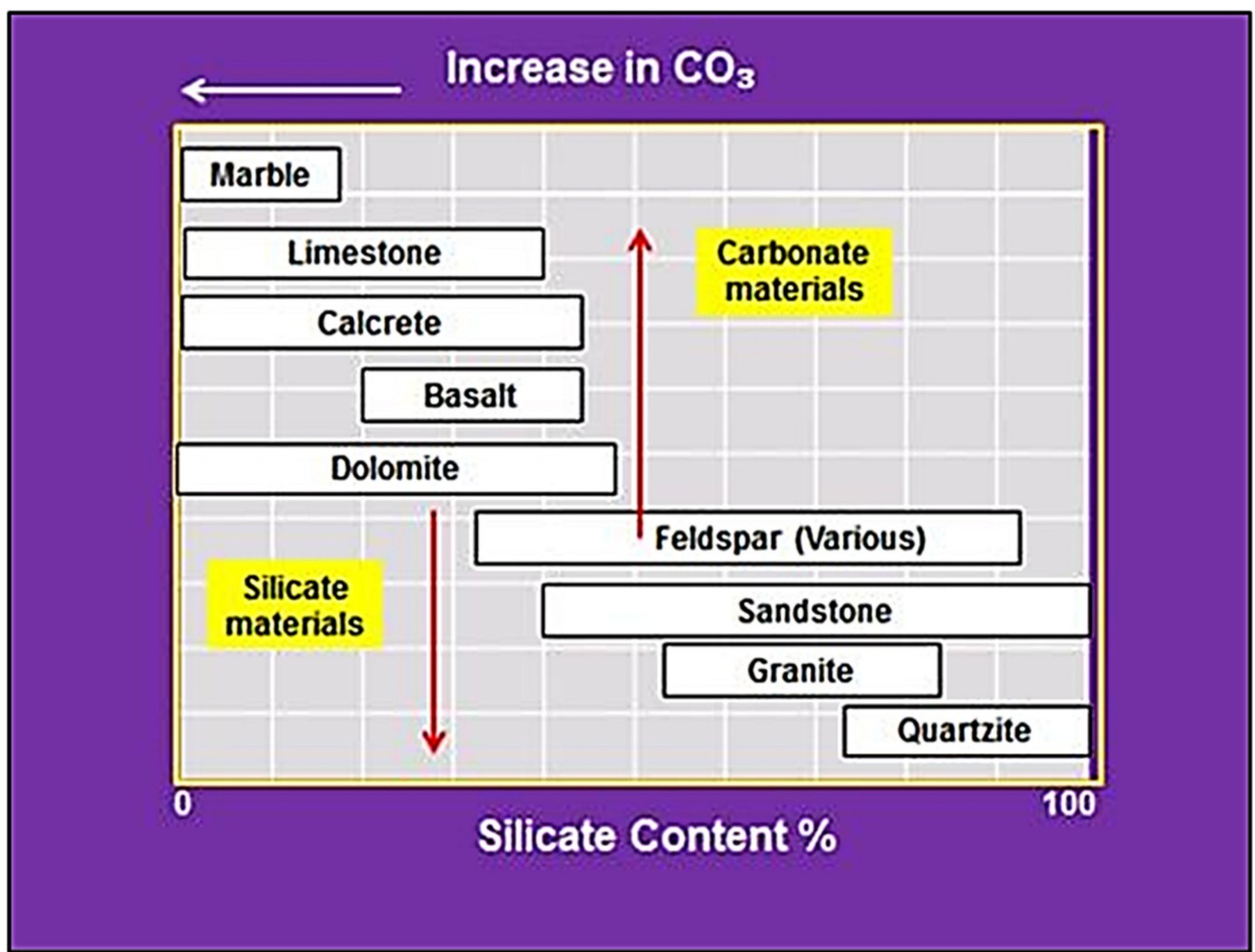

**Figure 3.** Silicon content variation versus Carbon-dioxide content found in some typical and commonly known naturally available materials.

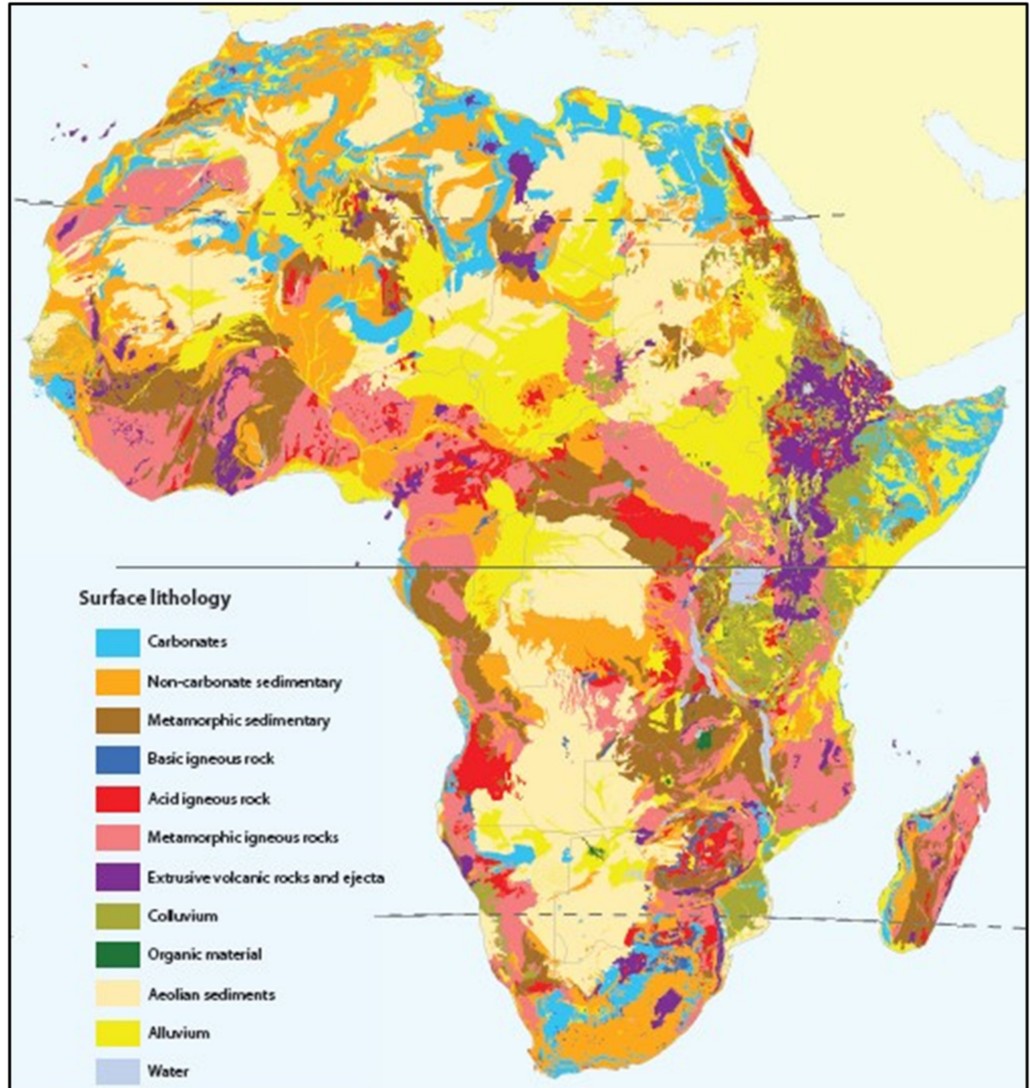

**Figure 4.** Ref. [32]: Example of the variation in surface geology of the continent of Africa showing large parts of the content that a general assumption of the presence of Silicate materials could have dire design and construction consequences.

The crust of the Earth is anything from $\pm 5$ km (oceanic) to $\pm 70$ km (continental high mountains) thick, and only the surfacing is subjected to the harsh climatic conditions conducive to chemical weathering and conditions resulting in sedimentary materials. Over and above the obvious, such as the non-Silicate materials mentioned, naturally available materials in climatic regions conducive to chemical weathering (Table 1) could have considerably reduced percentages of the Silicate materials due to a process of dilution or "washing out" in high rainfall areas, i.e., especially regions near and around the tropics.

The distinction between Silicate and non-Silicate and Silicon-poor materials is of importance as it forms the basis for the design and selection of an applicable effective NME stabilising agent. High percentages of other minerals such as Calcium Carbonates ($CaCO_3$) in the material to be stabilised will require a different modification in terms of a Hydroxy Conversion Treatment (HCT) [3,11] in order to provide high strength "permanent" bonding of the stabilising agent with the available elements in the material to meet engineering requirements.

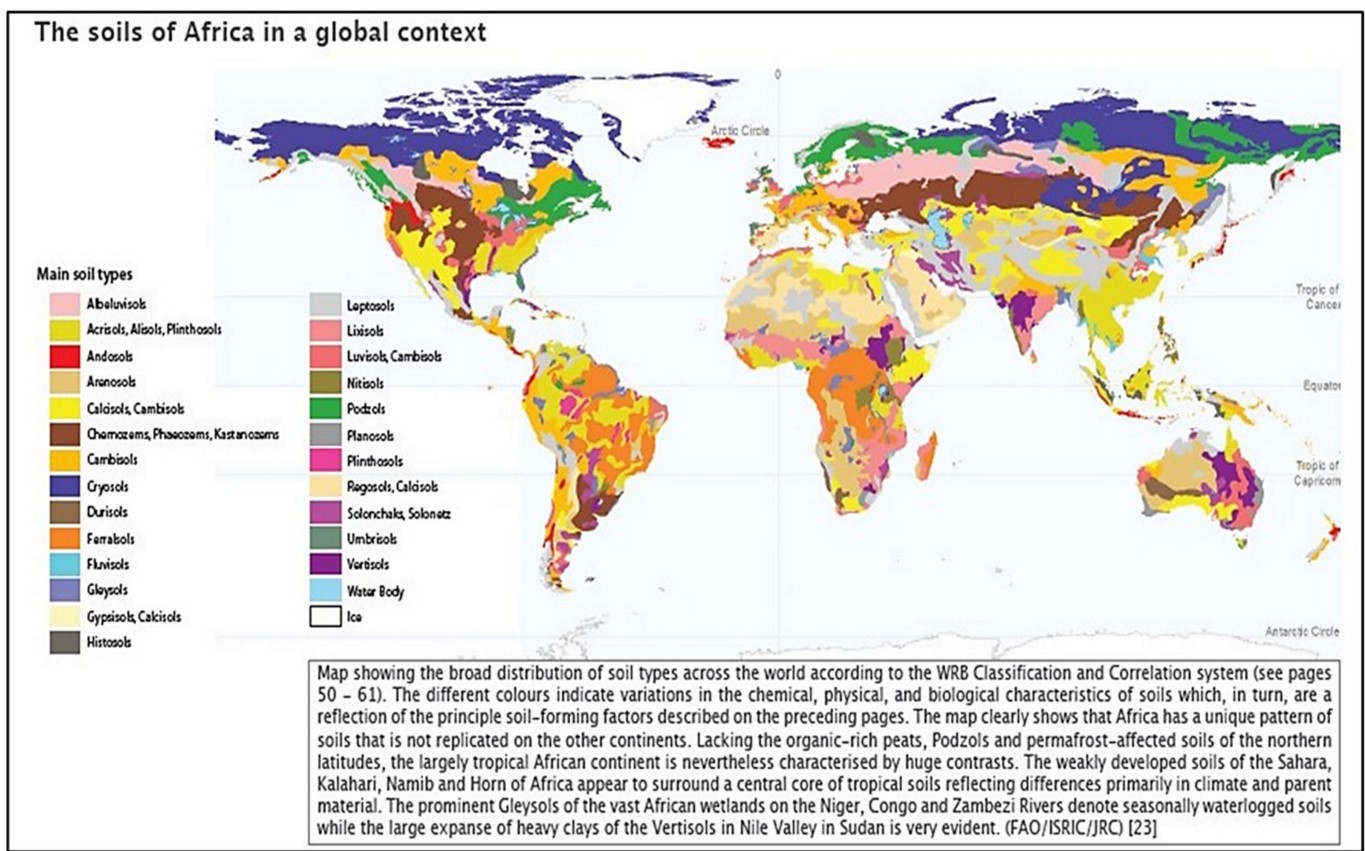

**Figure 5.** Ref. [32]: Distribution of surface material characteristics (soils) of the African continent in the global context with specific emphasis on the distribution of non-Silicate materials, such a Calcisols and Camisoles.

It should be noted that even Feldspars (which compromises about 60 per cent of the crust of the Earth (Figure 2)) may, in terms of the broad mineral classification of Alkali and Plagioclase (Figure 6 [33]), vary considerably depending on the basic elements present in the material. Plagioclase could contain high concentrations of free elements (elements with broken bonds available for effective bonding with a stabilising agent) such as Calcium (Ca) that will influence the classification and the selection of a material-compatible modification of an appropriate stabilising agent. Table 3 [33] gives an indication of the variation possible within the Plagioclase Feldspar group of materials, as indicated in Figure 6.

**Table 3.** Variation in minerals within the Plagioclase Feldspar group of materials as shown in Figure 6 [33].

| Mineral | % Albite (Na(AlSi$_3$O$_8$)) | % Anorthite (Ca(Al$_2$Si$_2$O$_8$) |
|---|---|---|
| **Albite** | 100–90% Ab | 0–10% An |
| **Oligoclase** | 90–70% Ab | 10–30% An |
| **Andesine** | 70–50% Ab | 30–50% An |
| **Labradorite** | 50–30% Ab | 50–70% An |
| **Bytownite** | 30–10% Ab | 70–90% An |
| **Anorthite** | 10–0% Ab | 90–100% An |

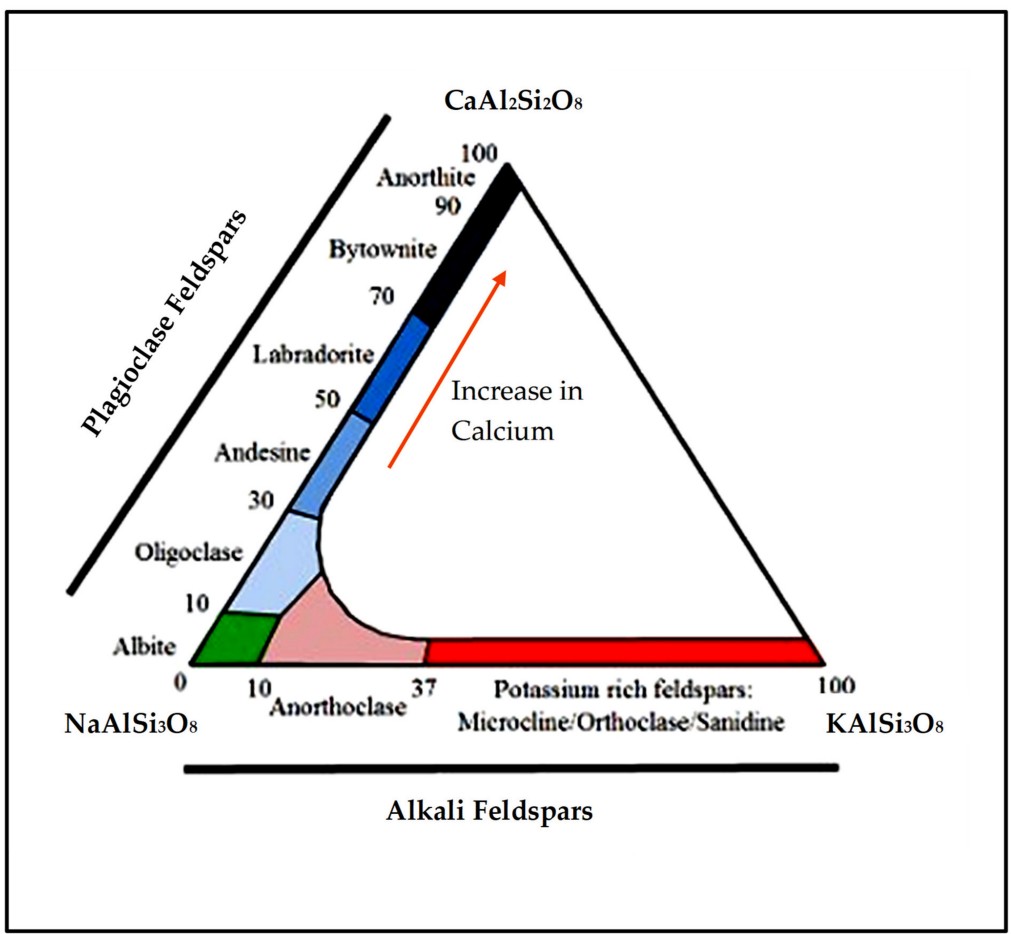

**Figure 6.** Various minerals that form part of the Feldspar group of materials [33].

The identification of the primary minerals comprising the materials could be pivotal to determine the success of any nano-modification of a stabilising agent. At a bare minimum, the percentages of the main elements, as identified in Table 2, must be determined. Experience has shown that basic assumptions, in terms of the general geological maps (in combination with normal material indicator testing), can result in "unexplained" expensive failures, occurring even during construction using traditional stabilising reactive stabilising agents. These are failures that could have been prevented through relatively inexpensive scientific testing, identifying the minerals present in the materials [10].

### 3.2.3. Secondary Minerals Defining the Condition of the Stone

The selection of the most effective material compatible NME stabilising agent is based on both the type and condition of the stone/aggregate/soil [3] for which it is intended to be used. Hence, material-specific designs are created based on the primary minerals present in the materials, as well as the secondary minerals that have developed through the ages due to chemical decomposition as a result of weathering under specific climatic conditions. Weathering patterns are closely related to climatic conditions, and climate plays an important role in the differences in the expected condition of the naturally available materials in different regions of the world as illustrated in Figure 7 [34]).

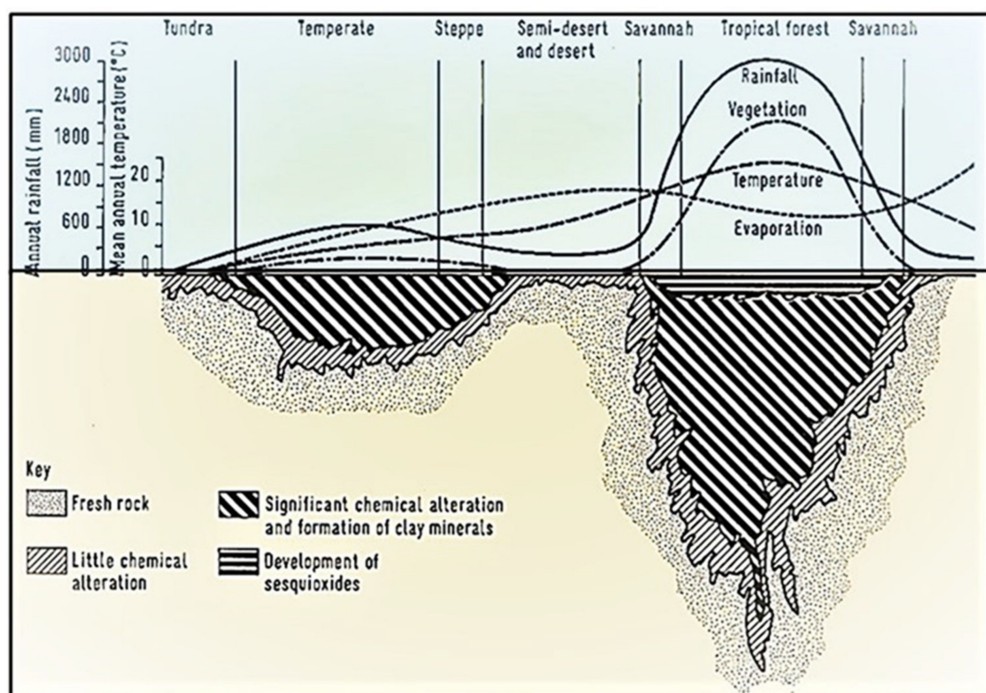

**Figure 7.** Illustration of the influence of various factors on the extent of chemical weathering of materials on the surface of the Earth (created from [34]).

A typical weathering pattern found under tropical conditions is illustrated in Figure 8 [35], showing a gradual change from fresh rock to a completely weathered soil at the top. The chemical weathering process from the basic rock formation (Figure 8) is illustrated in Figure 9 (recreated from [35,36]), showing the various stages of chemical transformations from the apparent rock and primary minerals to the formation of secondary minerals through various stages.

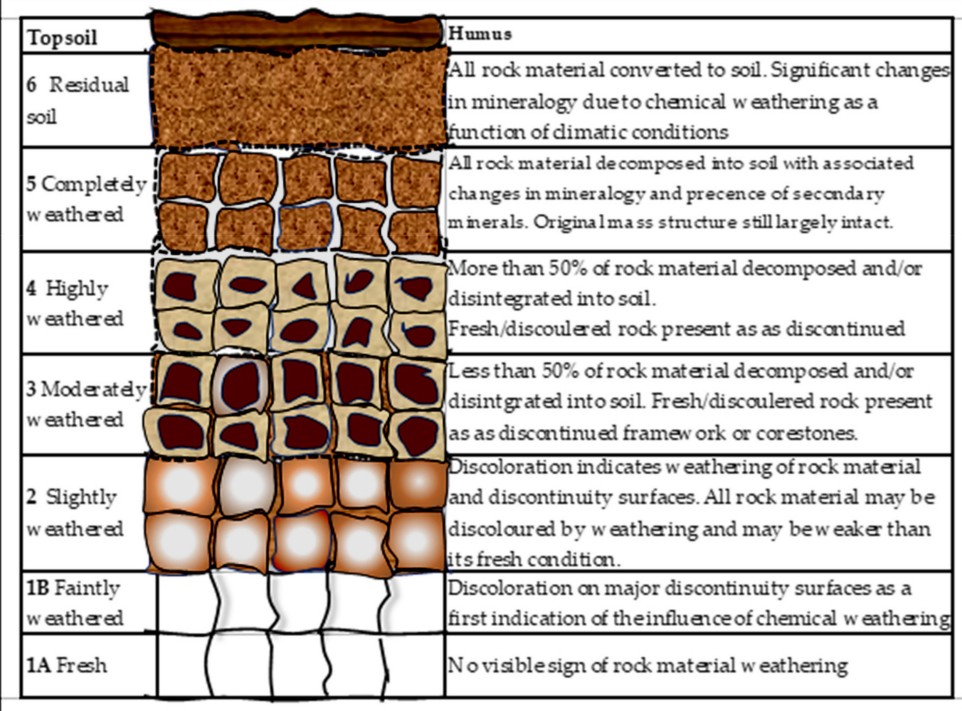

**Figure 8.** Schematic illustration of tropical weathering profiles (based on [35]).

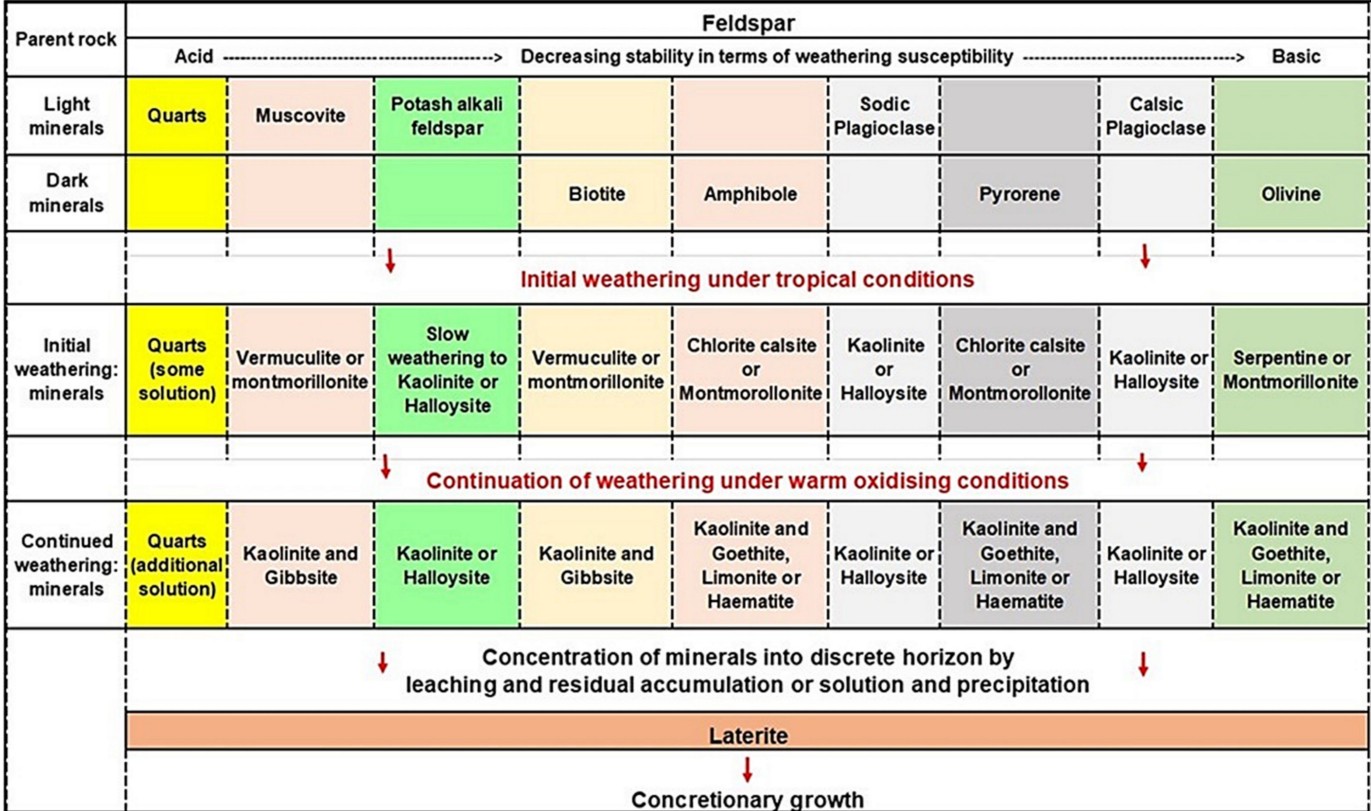

**Figure 9.** Chemical decomposition through weathering and the formation of secondary minerals through the transformation of primary rock formations (recreated from [36]).

For example, the distribution of one of the most frequently occurring naturally available materials found in the humid and tropical regions of the world is shown in Figure 10 [36,37], often referred to as "Laterite". Laterite is a term often misused, referring to basically any weathered granular materials/soils in colour varying from reddish-brown to brown or even yellow. Laterites also vary considerably in mineral composition, depending on the state of weathering as shown in Table 4 [28]. It is seen that Laterite and Laterite soil may contain anything from 5 to 70 per cent Silicon-dioxide ($SiO_2$), with similar variations in Ferric-Oxide ($Fe_2O_3$) and Aluminium-Oxide ($Al_2O_3$). It follows that Laterite is generally defined in terms of the sesquioxide ratio (S/R), where [38]:

$$S/R = [SiO_2/(AL_2O_3 + Fe_2O_3)] \tag{1}$$

where:

    S/R < 2 defined as Laterite soil;

    S/R < 1.33 defined as Laterite.

The engineering properties of the naturally available materials will change considerably over time, depending on the stage of chemical weathering (Figure 11 (recreated from [39])). As previously discussed, the mineral composition with specific reference to the silicon content, can have a considerable influence on the bond strengths achieved with the material when using a specific nano-modification of a stabilising agent [11]. Materials containing a low percentage of free silicon elements will require a stabilising agent modified with an additional HCT treatment to form chemical bonds of adequate strengths to ensure a long-term durability with available free elements within the naturally available materials. The high-strength chemical bonds will also ensure that the material maintains the hydrophobic level specified for the material in the specific pavement layer, as designed.

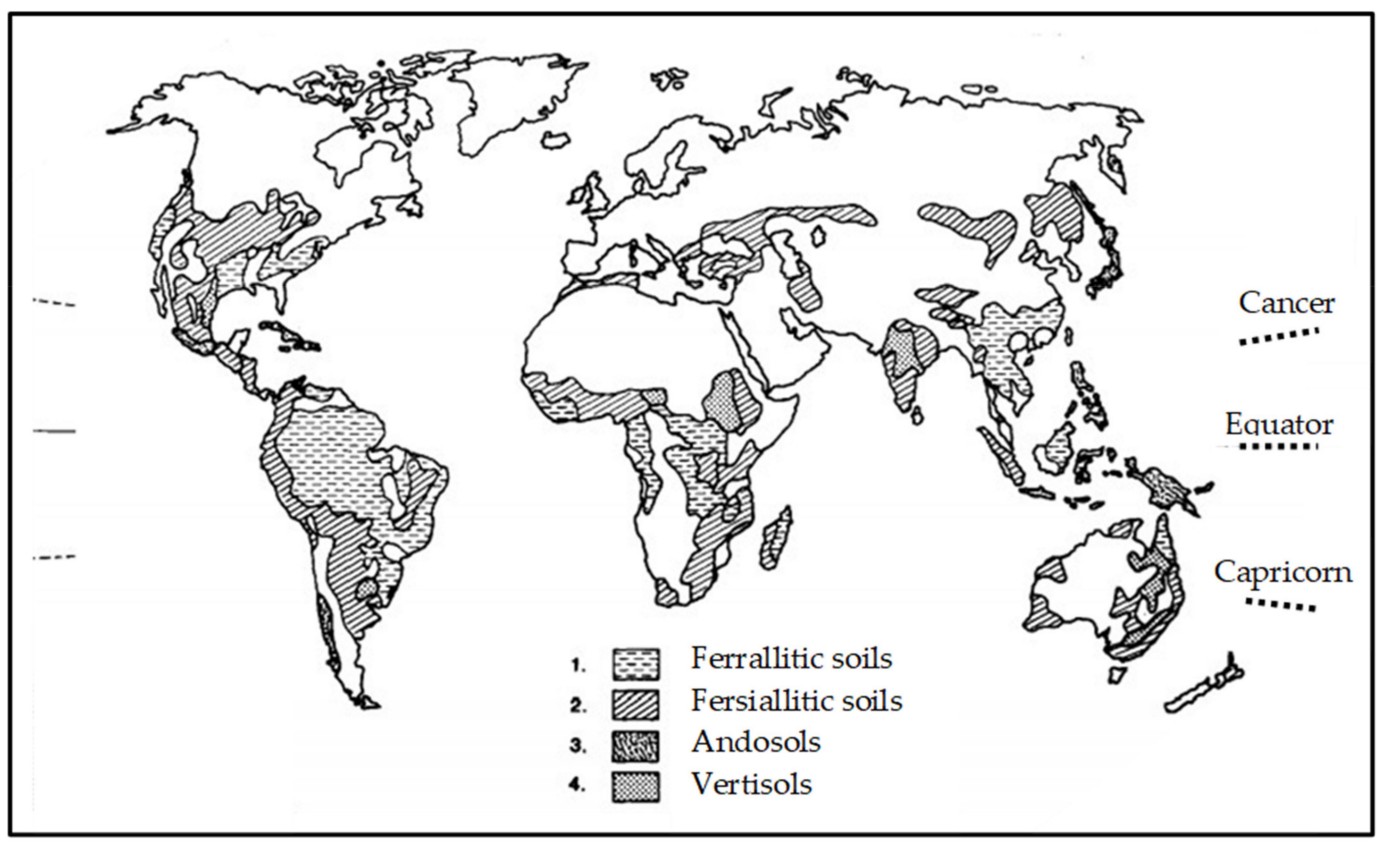

**Figure 10.** Refs. [36,37]: Distribution of the main soil types commonly referred to as "Laterites"—these softs extend beyond the tropics in favourable circumstances, including high-rainfall, sub-tropical, continental East coasts (Ferrallitic soits) and the West coast/Mediterranean and continental interiors in mid latitudes (Fersiallitic soits).

**Table 4.** Typical variation in mineralogy of Lateritic material [28].

| Component | % By Mass | Main Form of Occurrence |
|---|---|---|
| $SiO_2$ | 5–70 | Quartz, feldspar, clay minerals |
| $AL_2O_3$ | 5–35 | Feldspar, clay minerals, gibbsite |
| * $Fe_2O_3$ | 5–70 | Goethite, hematite |
| $TiO_2$ | 0–5 | Anatase, rutile |
| MnO | 0–5? | |
| $P_2O_5$ | 0–1 | |
| $H_2O+$ | 5–20 | Clay minerals, goethite, gibbsite |
| Loss of Ignition | 5–30 | Clay minerals, goethite, gibbsite, organic matter |
| Organic Matter | 0.2–2 | Organic matter |

Note: * Total iron on $Fe_2O_3$.

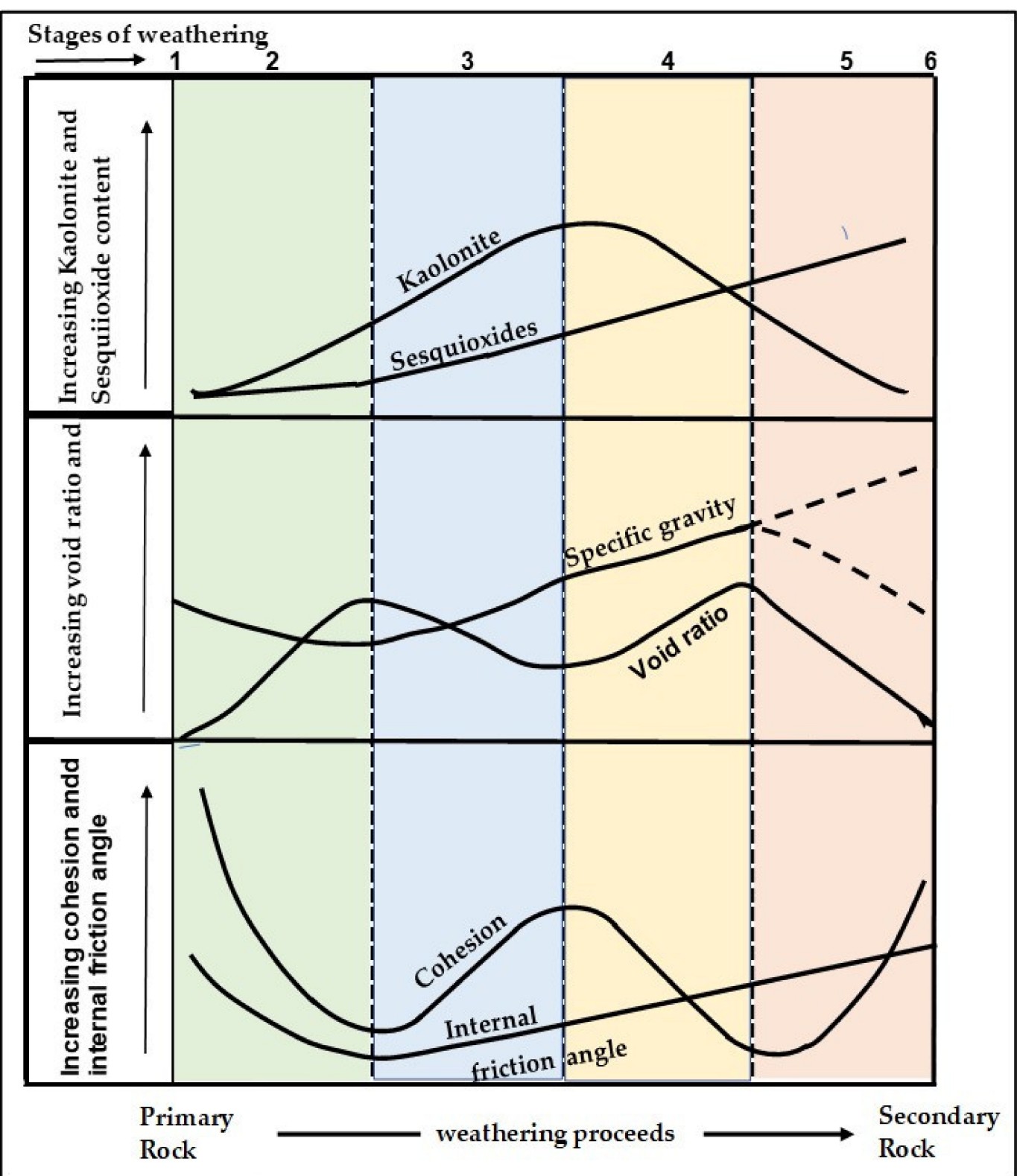

**Figure 11.** A model for variation in engineering properties of basalt derived lateritic soils resulting from weathering (Recreated from [39]).

It is seen that, with time and at advance stages of chemical weathering, more and more clay particles will form (e.g., Montmorillonite, Kaolinite, etc.). The increase in the clay content as part of the naturally available materials presents additional problems in terms of:

- The size of the molecule of the stabilising agent;
- The size of the particle of the nano-silane modification of the stabilising agent;
- The practical considerations during construction.

The "clay problem" is a subject that in itself requires extensive discussion, which will not be covered in any detail within this article. In order to appreciate the considerable challenges offered by materials containing a high percentage of clay, the properties of some of the most commonly found clays are summarized in Table 5 (based on [40]).

**Table 5.** Clay characteristics of importance (Based on [40]).

| Property | Type of Clay | | | | |
|---|---|---|---|---|---|
| | Smectite (Montmorillonite) | Vermiculite | Illite | Chlorite | Kaolinite |
| Specific Area ($m^2/g$) | 700–800 | 500–700 | 50–200 | 25–40 | 5–30 |
| External surface | Very High | Very High | Medium | Medium/low | Low |
| Basal spacing (Å) | 9.6–20 | 10–15 | 10 | 14 | 7.2 |
| Internal surface | Very High | High | Medium | Medium | None |
| Expanding | Yes | Slightly | No | No | No |
| Swelling capacity | High | Medium High | Medium | Low | Low |
| Cationic exchange capacity (CEC) (meq/100g) | 80–100 | 100–150+ | 10–40 | 10–40 | 3–15 |
| Similar clays | Beidellite Nontronite Saponite Bentonite | | Mica $Fe_2O_3$ $Fe(OH)_3$ $Al(OH)_3$ | | Halloysite Anauxite Dickite Nacrite |

The specific surface area of the different clays (in $m^2/g$) shown in Table 5 is of high importance when stabilising materials containing high percentages of clay using a nano-silane modifications and the selection of an applicable stabilising agent. In order to achieve a high level of hydrophobicity, the total area of all the particles in the material needs to be covered. It is seen that the specific surface area of clay could vary from about 700 to 800 $m^2/g$ (Smectite) to 7 to 30 $m^2/g$ (Kaolinite), a considerable difference that will determine the percentage of a nano-silane of appropriate size to be required to achieve a durable hydrophobic effect that will be lasting under the effect of loading. It follows that the smaller the particle size of the nano-silane, the less volume will be required to penetrate the clay matrix to achieve the required hydrophobic effect and vice-versa.

However, at the same time, the Cationic Exchange Capacity (CEC) of the specific clay is just as important. The higher the CEC, the more suspectable to chemical interaction the clay will be. Smectite, for example, has a comparatively very high specific area in combination with a high CEC. Hence, Smectite is much more susceptible to chemical alteration than, for example, Kaolinite, which has a relatively low specific area but a similarly low CEC (7 to 15 meg/100 g). A low CEC in combination with relatively small particle sizes may prove difficult to chemically alter and firmly bind together through any modification. In all definitions, Kaolinite can be considered as a "dead" clay with little attraction to nutrients and water for agricultural purposes. Similarly, materials containing relatively high percentages of Kaolinite present considerable challenges towards the effective strong formation of chemical bonds during the stabilisation process. The challenge is to penetrate the clay matrix, activate the surface of the clay with an applicable HCT and to create the chemical bonds that will enable the material to be stable and durable under loading conditions.

Clay crystals are usually in the order of 1 to 2 nm in size (Figure 12 [41]). These crystals form nano-plates that (depending on the type of clay) are susceptible to water "capturing", resulting in the considerable swelling of the material (e.g., the Smectite group of clays). It follows that in order to effectively utilise materials with a high clay content, specialised technologies are required to achieve the required hydrophobic nature, expel the water and achieve the strengths required through stabilisation. Clay in a dry state has a high bearing strength. The problem is that clay in nature is usually associated with water "capturing", and to make it hydrophobic, water is required as a carrier fluid for the nano-silane modifier to reach the clay particles. The clay particles will release and repel the water molecules only after the characteristics of the clay particles have been changed. After repelling the water particles, the material will have to reach a moisture content not exceeding the Optimum Moisture Content (OMC) of the modified material (in the order of 10 per cent less than the original OMC) before stabilisation, compaction and strength can be achieved. The process of the effective utilisation of materials containing a high clay content presents technological as well as practical challenges.

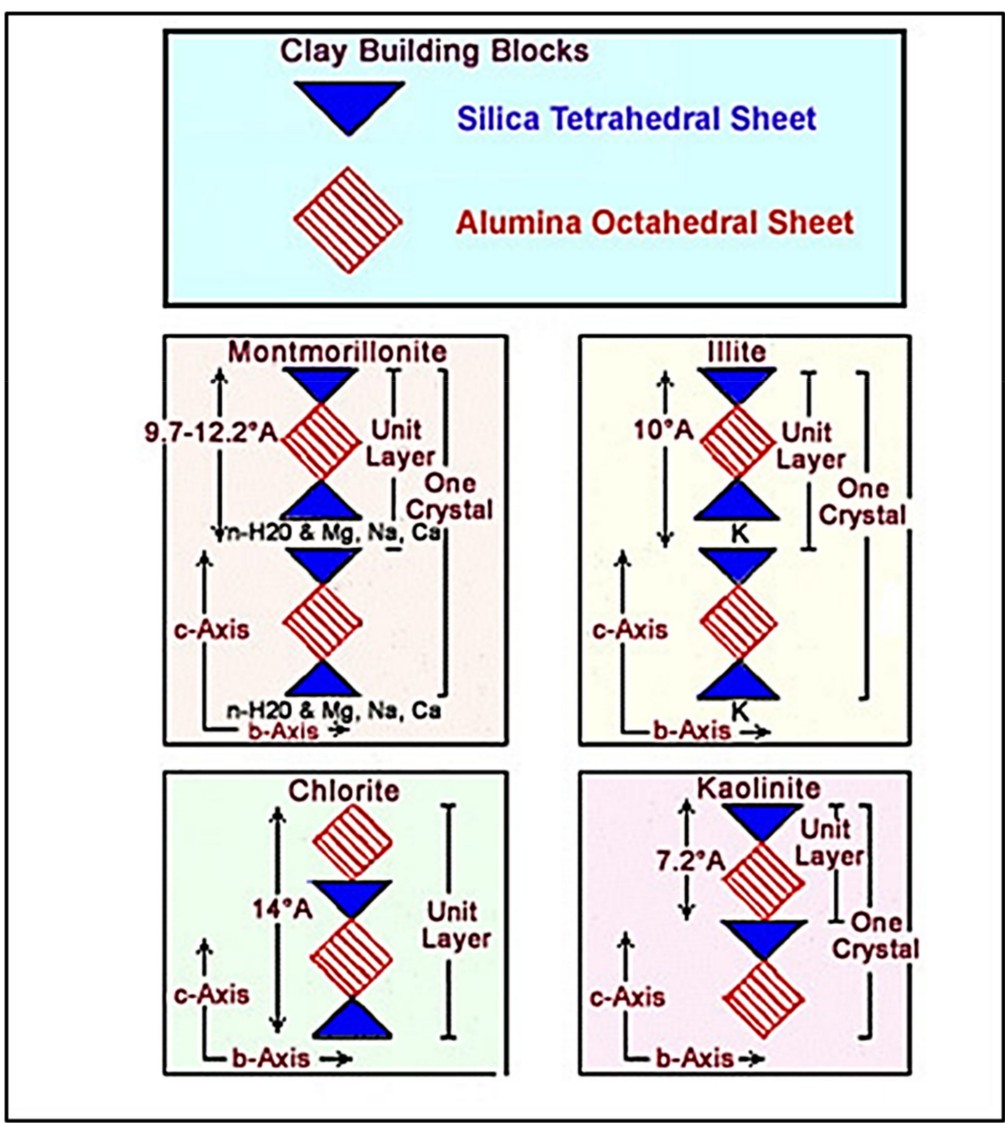

**Figure 12.** Ref. [41]: Some typical sizes of clay crystals as measured in Angstrom (where 1 Angstrom (Å) = 0.1 nm).

Generally available nano-silane particles are in the order of 2 to 5 nm in size and will be ineffective to penetrate the clay matrix and render clay particles hydrophobic.

The treatment of several clay nano-layers is easily achievable using these technologies. However, any stress/strain put on these samples causes slip-planes within the layers, which allow water to have access to the clay nano-layers that have not been accessed and have not been chemically changed to become hydrophobic. The result is that the material becomes hydrophilic, absorbs water, swells and breaks any bonds created by an applied stabilising agent.

Ideally, the area of the particles to be bound together should be matched by the area that can be covered by the stabilising agent. With materials containing a grading with a high percentage of clay (<0.002 mm in size), the stabilising agent should have a grading to match the increase in the area to be covered, creating a stable environment within the material. This concept is illustrated in Figure 13, the practical implementation of which is discussed in the following recommended detailed material design method.

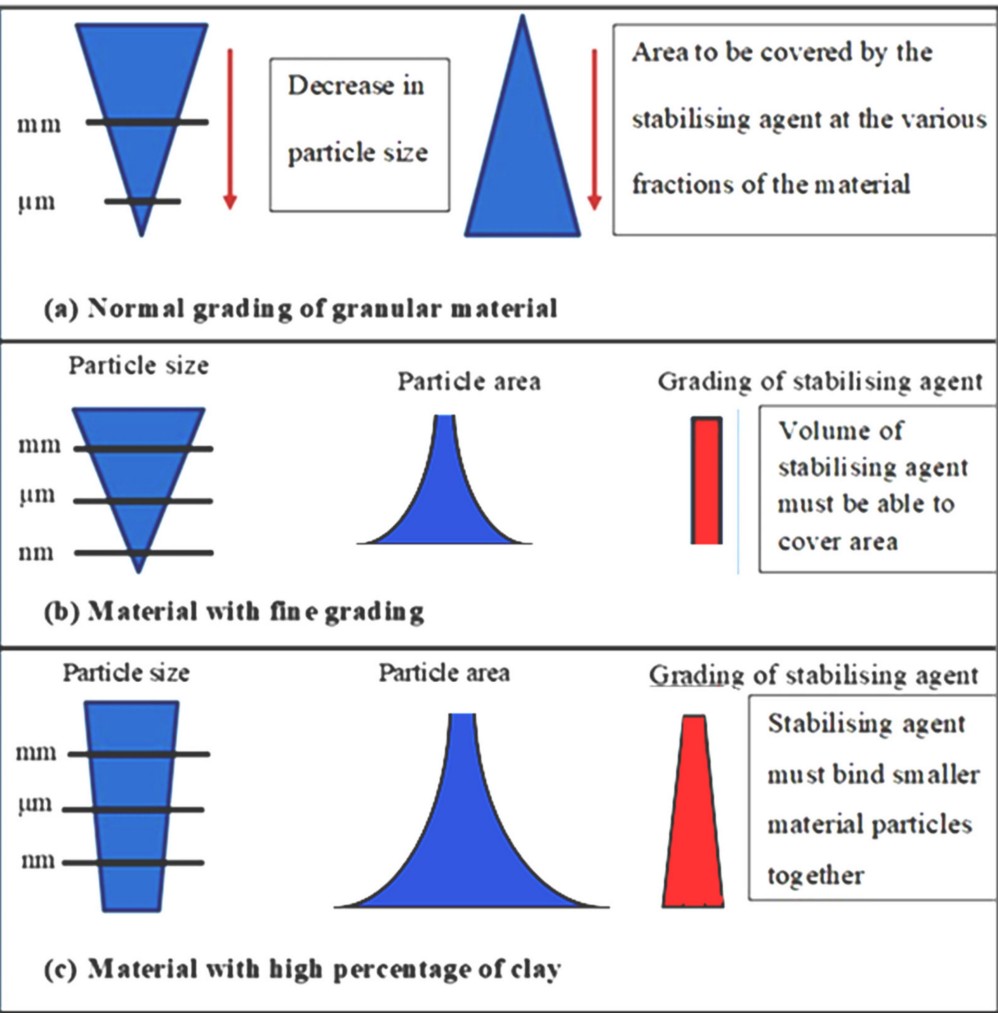

**Figure 13.** Illustration of grading of material (particle size) versus area to be covered by an applicable stabilising agent and required particle size of the stabilising agent.

Hence, to effectively treat materials containing high percentages of clay, the nano-silane particle must be of the highest quality and purity of a sub-nano (Pico) size in order to penetrate the clay nano-matrix. Only nano-silane particles exhibiting these characteristics will be able to replace the captured water molecules by binding with free cations present in the clay particles. By penetrating the clay matrix, the pico-size silane can render each clay nano-layer hydrophobic, repelling the captured water molecules and preventing any water from penetrating the clay nano-layers. No shearing of the clay nano-layers under loading will now expose the clay matrix to allow water ingress to occur, and a durable

stable layer will be created. This process will result in a considerable reduction in volume when the water is repelled from the clay nano-layers, which will normally (in the case of high-swelling clays) be associated with considerable shrinkage, as shown in Figure 14.

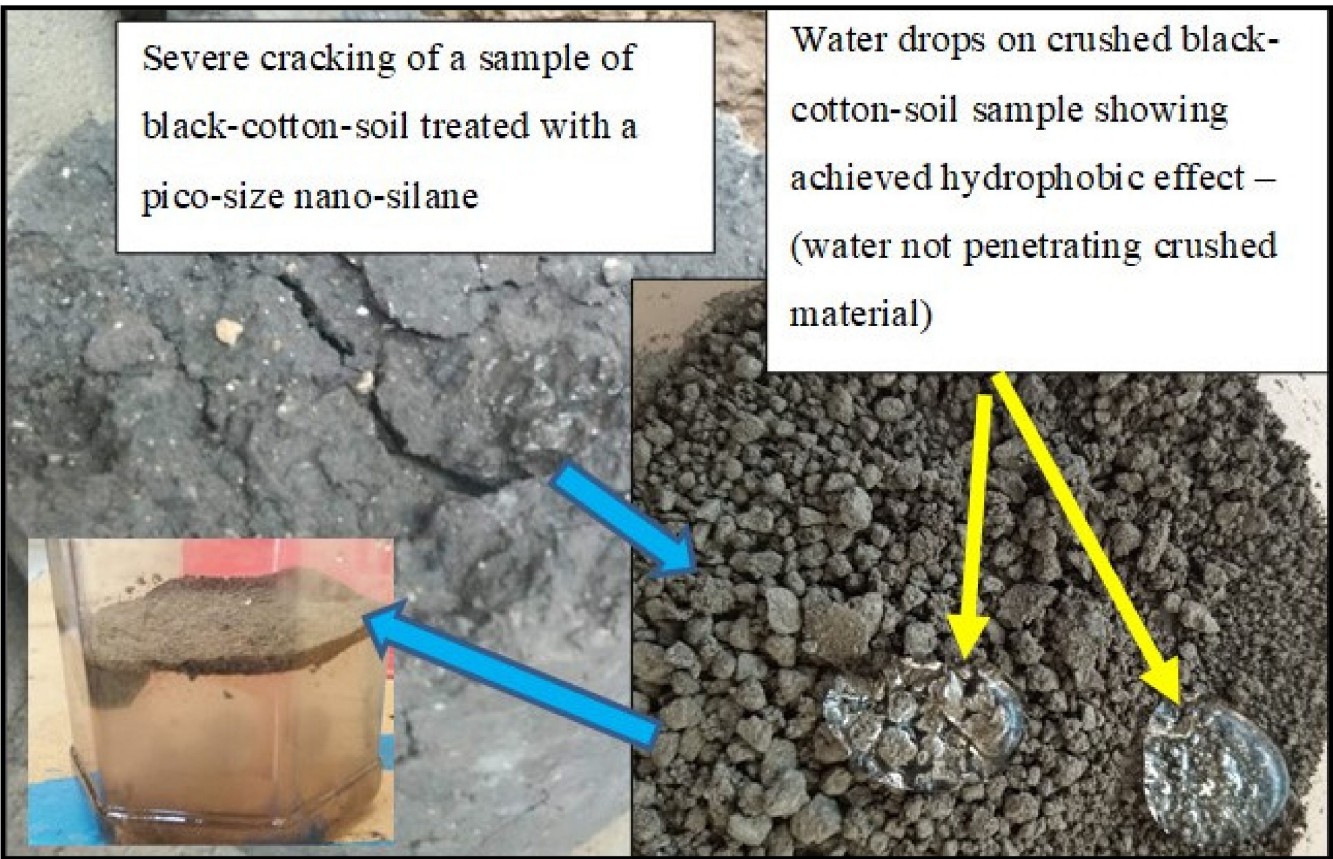

**Figure 14.** Treatment of black-cotton soil to obtain hydrophobicity with crushed clay particles floating on water (tests performed at GeoNANO Technologies, 2020—refer Acknowledgements).

Knowing the properties of the materials, a hydrophobic clay matrix can be achieved with an adequate amount of applicable nano-silane of sub-nano particle size (even for materials commonly referred to as black-cotton soils) in a laboratory environment. The stabilising agent used in conjunction with the nano-silane will also have to be able to actively bind clay particles together. Micro-size stabilising agent particles (1000 times bigger than a nano-size particle) will be too large and not effective as a stabilising agent, with the small size clay particles, in a practical sense, "swimming" within the stabilising agent.

The normal size of a bitumen particle within a bitumen emulsion or NME varies between 2 and 5 μm and will obviously not be suitable for the stabilisation of materials containing a high percentage of clay. Similarly, most of the normally available polymers also have particle sizes of similar dimensions. However, high-quality, high-end, graded nano-polymers are available that can provide solutions to the treatment of materials containing high percentages of clays. Similar to the illustration shown in Figure 14, the material-compatible stabilising agent particles need to be small enough (preferably containing different particle sizes) to bind the material particles together and form a solid stabilised layer conforming to the required engineering properties while retaining a hydrophobic nature under loading.

The technologies are available and have been proven under laboratory conditions, as shown in Figure 15. The clay particles can be stabilised and remain in a coherent unit, even under severe conditions. Figure 15 shows a sample of black-cotton soil treated

with a sub-nano-silane and stabilised with a Nano-silane Modified Nano-Polymer Emulsion (NMNPE), compacted by hand (still relatively loose). The water-drop on the top of the sample (Figure 15a) shows the perfect hydrophobic "beading" effect achieved (refer Figure 16 [12]). The same sample is shown in Figure 15b after being submerged in water for 7 days.

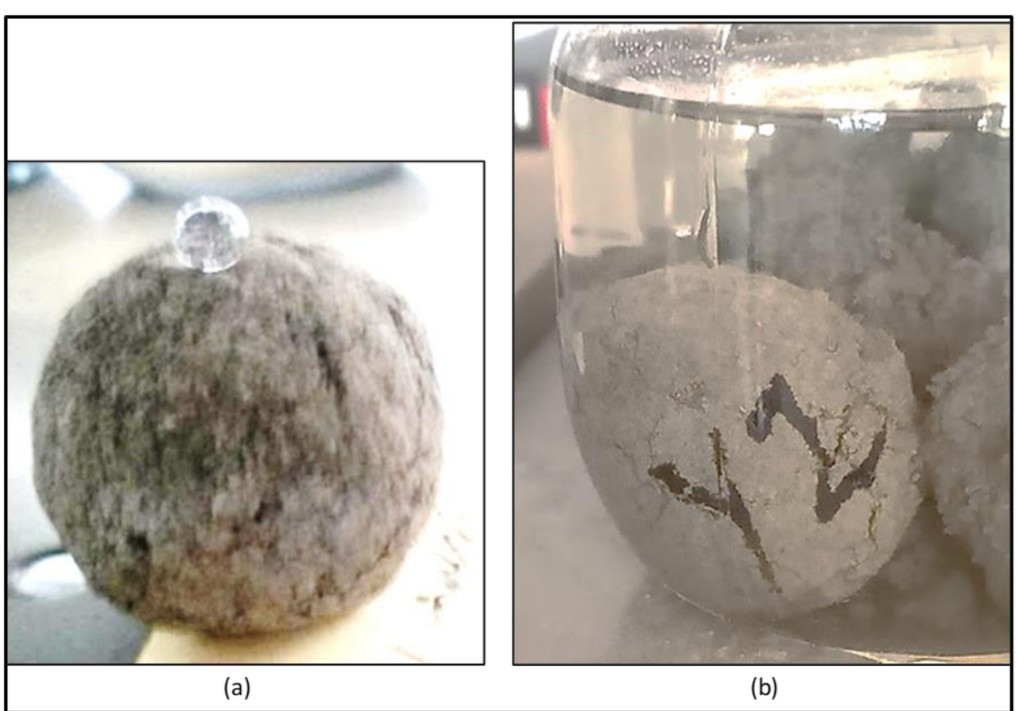

**Figure 15.** Black-cotton soil treated with a sub-nano-size silane and stabilised with a nano-silane-modified nano-polymer in an emulsion form (NME) compacted by hand with a water drop showing the perfect hydrophobicity achieved (**a**) and after 7 days submerged in water (**b**) (Testing performed at GeoNANO Technologies, 2020—refer Acknowledgements).

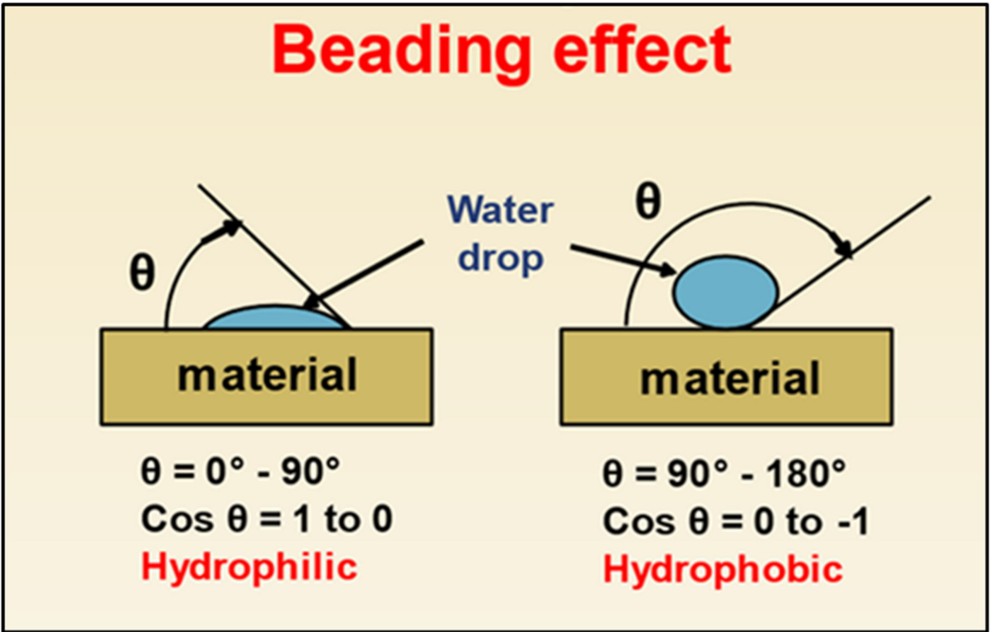

**Figure 16.** Ref. [12]: Illustration of the effect of hydrophilic versus hydrophobic effect.

The black-cotton soil represents some of the worst materials available and is used as a demonstration of the ability of new-age nanotechnologies that can find application in the field of pavement materials engineering. The practical application of these technologies under extreme climatic conditions with saturated conditions is a challenge that needs to be addressed in practice.

### 3.3. Identification of Scientifically Abased Material Characteristics

From the preceding discussions, it follows that some specific fundamental material properties normally measured are also of importance to the successful selection and application of a material compatible NME stabilising agent. These, inter alia, include:

- Maximum Dry Density (MDD);
- Optimum Moisture Content (OMC);
- Grading of the material properties. Of particular importance is the percentage of the material passing the 0.075 mm sieve and the grading of these fine particles. Hydrometer tests (not normally available in field laboratories) will give an indication of the silt and clay fractions. However, these tests will give no indication of the mineralogy or the type of clay present in the material.

For an increase in confidence and the limitation of risks, it is recommended that XRD tests [4,6,10] be performed as an input into the design process. Test protocols have been developed and are discussed in detail in the references and not repeated again [4,10]. The results from the recommended XRD scan test protocols of the materials will provide the designer with the information required to proceed with confidence to the next phase of the design process without following a trail-and-error, traditional pavement engineering approach.

### 3.4. Identification of a Material Compatible NME Stabilising Agent

The availability of the basic material information will enable a relatively easy design process to be followed to select a:

- Material-compatible applicable nano-silane modifier of a volume and dimension suitable to ensure that the required hydrophobicity is achieved, while also providing strong chemical bonds between the material particles and the stabilising agent;
- Compatible, applicable stabilising agent to be selected with the required grading to ensure that all material particles are bonded together in a stable, durable material, meeting the fundamental engineering requirements/criteria/specifications.

XRD scans will provide the information required and identify the primary and secondary minerals present in the naturally available materials as discussed [4,10]. Together with the physical material particle size measurements, in particular, information on the fraction passing the 0.075 mm sieve and the fraction passing the 0.002 mm size (clay, talk and mica), will determine the applicable nano-silane technology to be selected and the quantity thereof to be applied for laboratory verification.

A flow diagram of the required material information to determine a material compatible NME stabilising agent is shown in Figure 17, the use of which is demonstrated in Figure 18. Although CBR is considered not of importance with regard to the selection of an applicable nanotechnology solution, it is at this stage incorporated into the design figures as a reference for practitioners to traditionally used test methods.

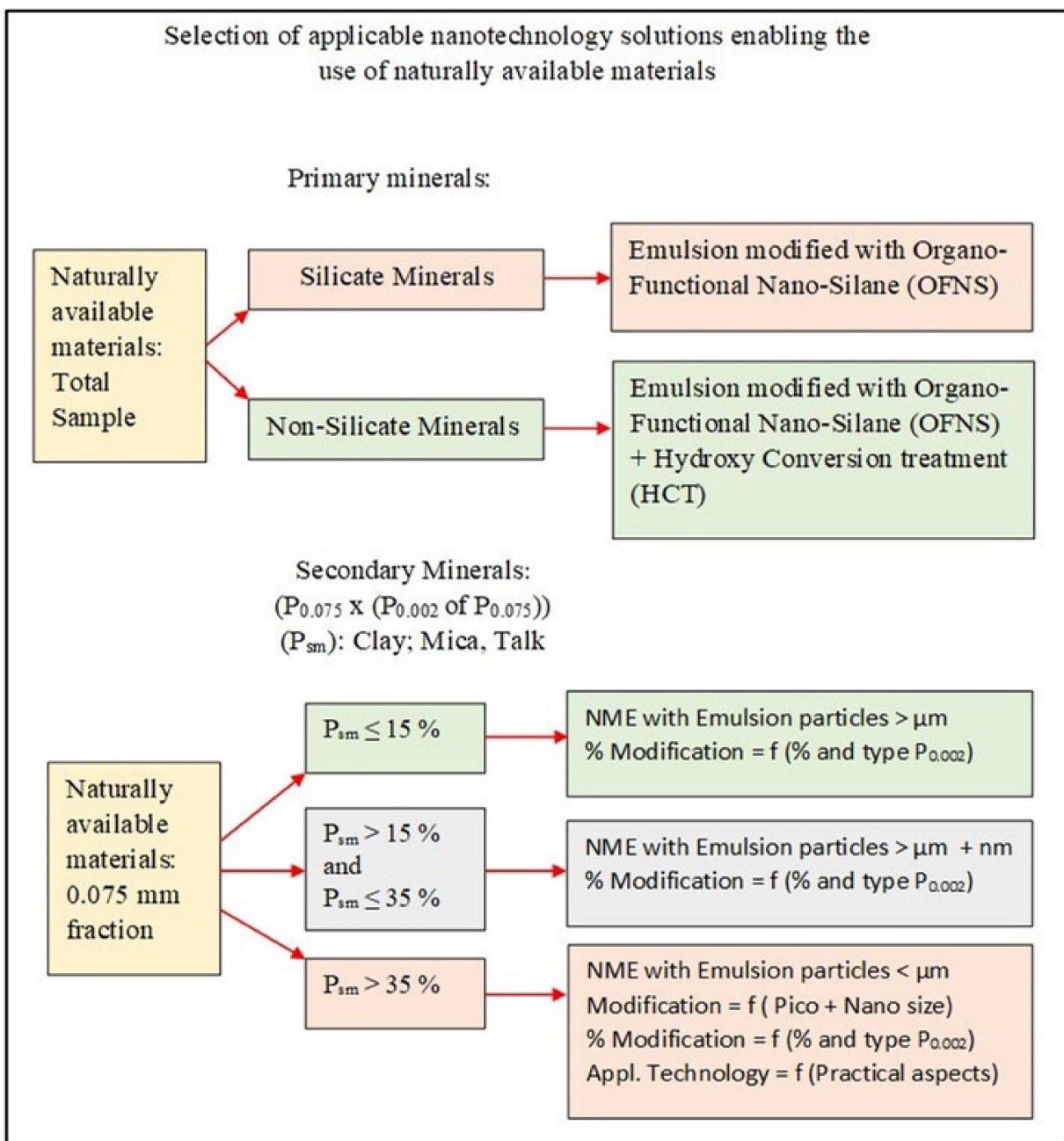

**Figure 17.** Required information as an input to the selection of a material-compatible Nano-Modified Emulsion (NME).

The surface area of the material to be covered by the material compatible NME stabilising agent is a function of the primary and secondary minerals present in the materials. An adjustment is made in the percentage of the material passing the 0.075 mm sieve size, taking into account the primary minerals and the comparable surface area shown in Table 6 (adjusted from [42]). Figure 17 (First adjustment) is based on Silicate materials. For non-Silicate materials, the percentage passing the 0.075 mm sieve needs to be adjusted on a pro-rata scale using the information given in Table 6 (adjusted from [42]). The clay, mica (muscovite) and talc information are used in the Second Adjustment in Figure 17, also considering the information contained in Table 5 for a comparative indication of the volume of a sub-nano-silane treatment required for materials with a clay, mica (muscovite) and talc content in excess of 35 per cent.

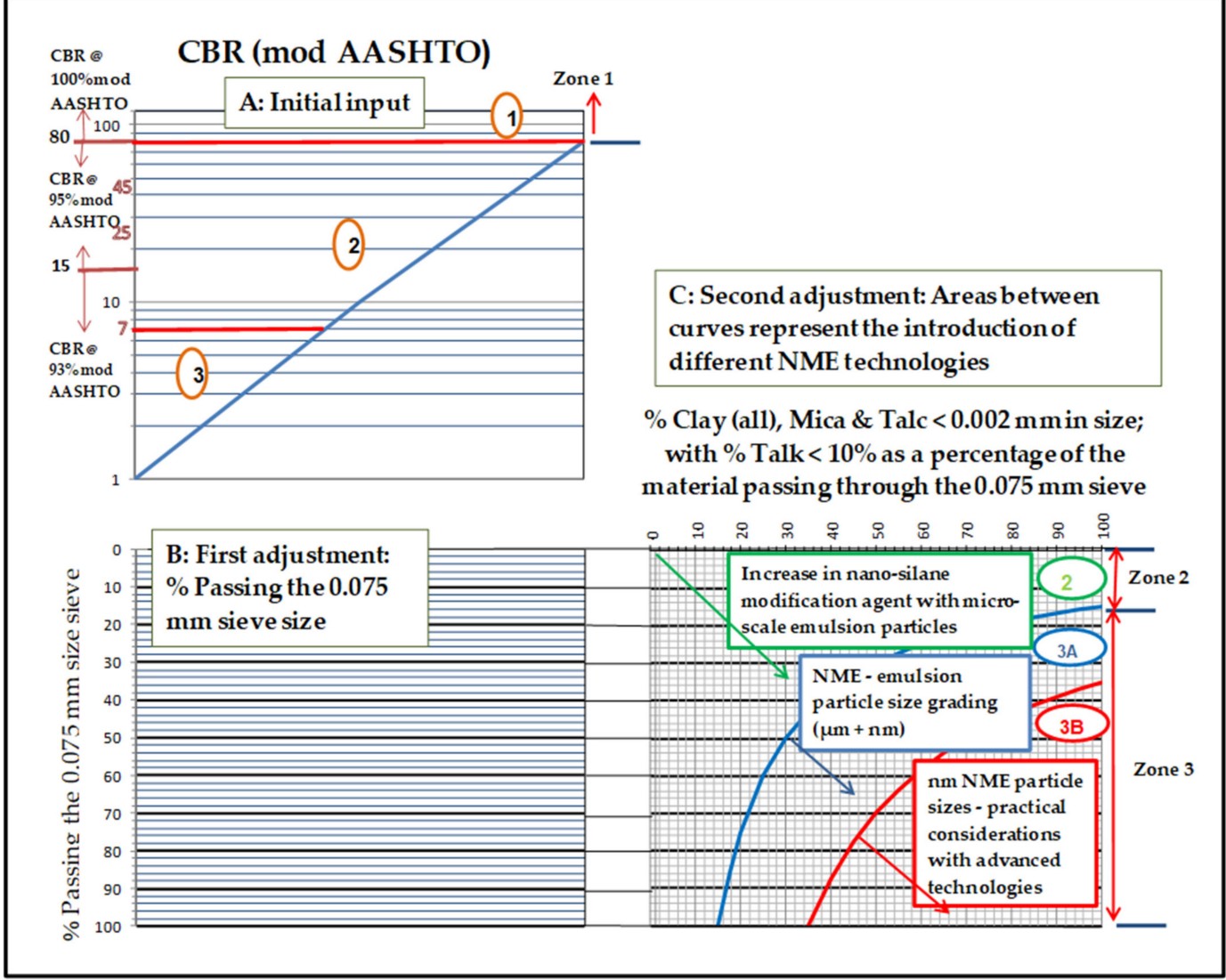

**Figure 18.** Demonstration of material input information required for the selection of a material compatible NME stabilising agent.

**Table 6.** Average relative surface area (multiplication factor) of different materials per unit mass ($m^2$/g) (calculated from [42]).

| Material | Surface Area of Material ($m^2$/g) | Comparative Surface Area with Relation to Silicate Materials |
|---|---|---|
| Glass | 0.1 to 0.2 | |
| Quartz ($SiO_2$) | 1 to 2 | 1 |
| Calcium silicate ($Ca_2O_4Si$) | 2.6 | 1.3 |
| Calcium carbonate ($CaCO_3$) | 5 | 2.5 |

The material properties in Zone 1 in Figure 18 are applicable to materials in pavement layers consisting of high-quality crushed stone. Zone 1 NME agents will protect the material from weathering due to chemical decomposition. This is achieved through the introduction of a water-repellent, material-compatible nano-silane agent together with a small percentage of a stabilising agent (e.g., bitumen emulsion) acting as a lubricant to assist in the compaction of the material and to more easily achieve the required high specified densities (102% mod AASHTO [22,23] or 88% of the Apparent Relative Density (ARD)) with considerably less energy input, effectively reducing the time of construction.

The complexity of the mineral composition of the materials in Zone 3B (Figure 18), requires a more detailed analysis due to the complexity of the mineralogy in the material as well as the practical implementation considering the high clay, mica (muscovite) and talc content (and hence, workability) of the materials. The use of Figure 18 in the selection of the most applicable type and relative volume of the NME stabilising agent and the applicable particle sizes of an applicable stabilising agent is demonstrated in Figure 19 through the use of an example.

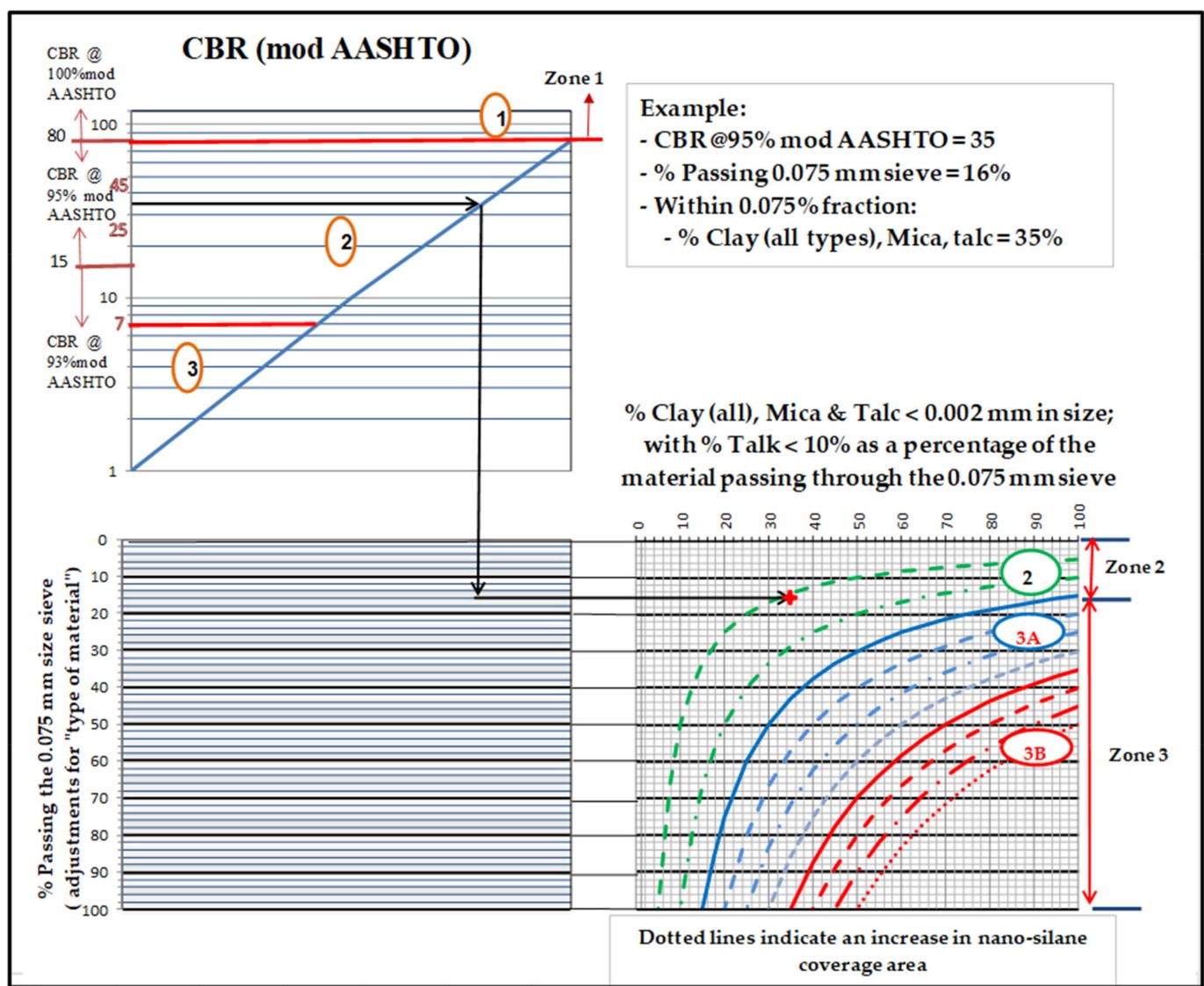

**Figure 19.** Example of the practical use of the material properties for the selection of an appropriate NME technology for the naturally available material.

The input material properties available are as follows:

- Primary mineral: Silicate material;
- CBR @ 95% mod AASHTO = 35;
- Percentage passing the 0.075 mm sieve = 16%;
- Percentage clay, mica and talc as a fraction passing the 0.075 mm sieve = 30%.

The example in Figure 19 shows that the material can be treated adequately with a normal NME with particle sizes of the stabilising agent >1 μm in size (e.g., bitumen emulsion or equivalent) modified with a relative volume of 2X of a material compatible nano-silane (with X being defined as the original required minimum area coverage).

Should the primary mineral in the example be calcretes, the effective % passing through the 0.075 mm sieve will be multiplied by a factor of 2.5 (Table 6) to an effective 40 (16 × 2.5) per cent. The modifying agent will require a HCT additive with a resultant change in the material compatible NME, as well as the volume of the nano-silane modifier used in the NME, as indicated by the adjustments if Table 6.

Figure 19 can be reduced in practice to:

- Figure 20, giving a relative increase in nano-silane modification in terms of the required coverage area as indicated by the example using a combination of nano-silane (X) and sub-nano-size (Pico) silane (Y) to cover the increase in area presented by the decrease in material particle size;
- Figure 21 as an indication of the applicable stabilising agent in terms of a micro-size and nano-size stabilising agent.

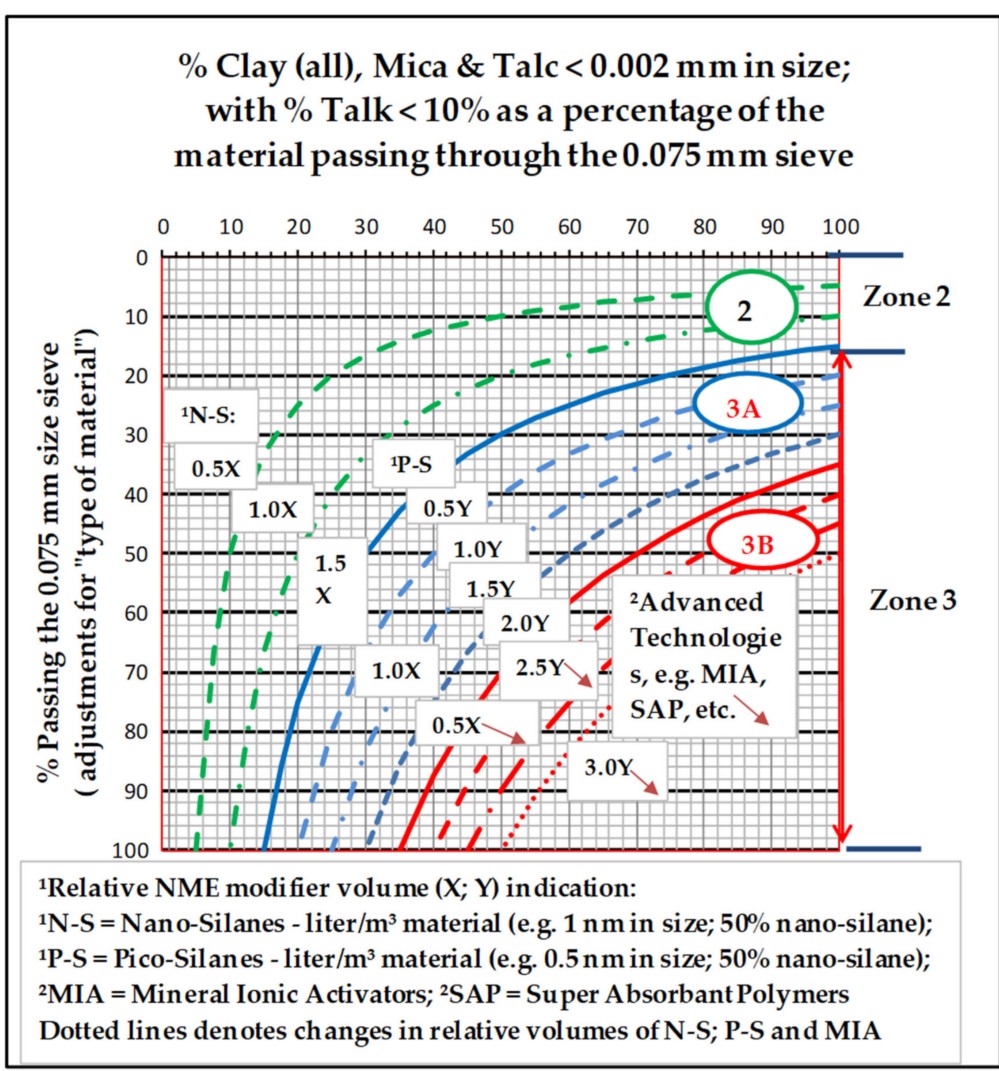

**Figure 20.** Selection of relative volumes of nano-silane modifications and high-quality, high-purity pico-silane modifying agents required to address the surface area of the naturally available materials.

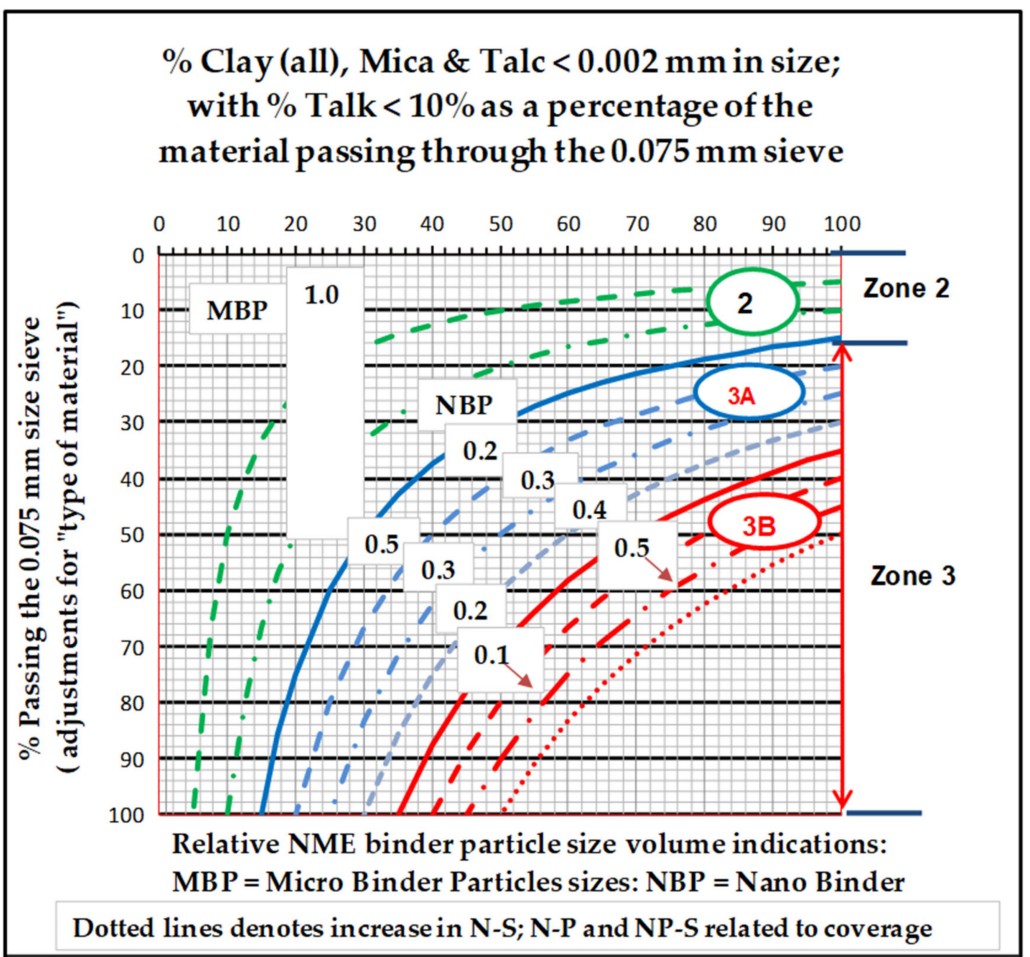

**Figure 21.** Selection of an applicable stabilising agent to be used in conjunction with the selected modifying agent (NME) to effectively stabilise the material and meet the required engineering properties in terms of stresses, strains and durability.

The use of these design figures is dependent on the size of the nano-silane particles and are relative indications influenced by the fine fractions of the available materials. The nano-silane particles assumed in this example are typically 1 nm and 0.5 nm in size. If nano-silanes of different dimensions are used, the necessary adjustments need to be made. The rest of the grading of the available materials has been found not to have a significant influence on the end result [10].

The most applicable percentage of the NME stabilising agent to apply is determined through a series of laboratory tests during the material design process, applying a variation in the percentage (per mass) of the NME stabilising agent to a representative sample of the naturally available material to be stabilised.

The following ranges in terms of the percentages of the NME stabilising agent (in terms of the mass of the material to be stabilised) are recommended to be tested in a laboratory during the design phase to determine the optimum percentage to be applied to the material to meet the required engineering properties most cost-effectively. These percentages are to be incorporated into the Bill of Quantities (BOQ) for a specific project (Zones refer to Figures 14 and 15):

- Zone 1: 0.1% to 0.2% per mass;
- Zone 2: 0.7% to 1.5% per mass;
- Zone 3A: 0.4% to 1.0% per mass;
- Zone 3B: Specialist design.

The use of relatively reduced percentages of the NME stabilising agent is possible due to the reduced particle sizes achieved through the introduction of the modified emulsification agent. These smaller particle sizes distribute with ease within the construction water, requiring no more effort than the normal construction of materials at a moisture content of about 10 per cent less than OMC.

Materials with a high percentage of fines may present additional problems in terms of practical construction challenges in the field. Some indication of materials qualifying for special construction practices are demarcated in Figure 22 (i.e., Figures 20 and 21 plotted on a log-log scale).

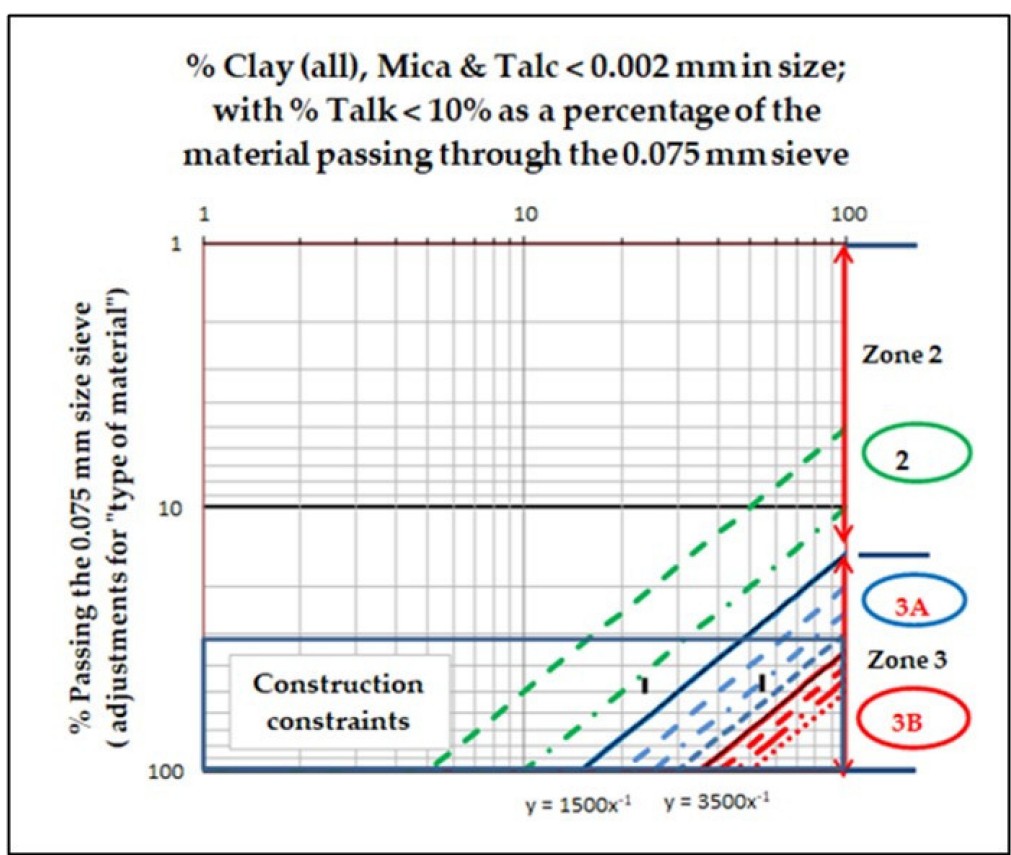

**Figure 22.** Materials with a high percentage of fines as demarcated, indicating materials which may present challenges in terms of construction practices in the field.

### 3.5. Required Fundamental Engineering Properties

3.5.1. Stresses, Strains and Durability

The engineering requirements depend on the design estimated traffic loading over the design period and the required bearing capacity of the pavement structure. In terms of Mechanistic-Empirical design approaches these engineering properties are defined in terms of the stresses and strains within each pavement layer in the pavement structure. In line with these modern analysis and design methods, tests are recommended to be performed to indicate the compressive and tensile strengths of the stabilised material. The criteria for the various structural layers are defined in terms of generally available test equipment indicative of the compressive and tensile strengths that can routinely and cost-effectively be performed in design laboratories as well as for quality control laboratories during construction. The Unconfined Compressive Strength (UCS) and the Indirect Tensile Strength (ITS) are recommended for use to evaluate the NME stabilised materials, in line with industry recommendations in place for Bituminous Stabilised Materials (BSM) (Bitumen Emulsion and Foam stabilised materials) [43].

In order to prevent confusion between the testing and evaluation of various stabilising agents used in the roads industry, technical terms and tests procedures are standardised as much as possible. However, in contrast to BSM designs, NME stabilised materials contain no cement additive, and samples for testing are consequently not placed in plastic bags, which is needed to assist with the hydration of the cement [13]. The testing of numerous naturally available materials throughout southern Africa have also shown results exceeding by some considerable margins, results usually obtained based on BSM designs. This aspect is to be expected due to the introduction of the organofunctional nano-silane modifier that acts, inter alia, as an aggregate adhesive. In addition, the particle surfaces within the material that is being stabilised are chemically altered to become hydrophobic, rendering the individual material particles to be lipophilic (oil/fat loving), not allowing water molecules to have access to the material minerals within the materials and hence, improving long-term durability.

Hence, an additional material classification is added in terms of retained compressive and tensile strengths to assess the hydrophobic resistance that is achieved with the NME stabilisation. This is done by comparing the test results after a rapid curing process ($UCS_{dry}$; $ITS_{dry}$) and after being soaked in water for 4 h ($UCS_{wet}$; $ITS_{wet}$) [13]. A high dry result associated with a low percentage of retained strength indicates that the water is able to affect the material strength properties. It follows that the volume of the nano-silane modifier as part of the NME stabilising agent, needs to be increased for the specific material to achieve the specified level of hydrophobic action required. Not only is the Retained UCS (RCS) and ITS (RTS) specified, but the results are also compared to the minimum specifications to assess the level to which the specifications are being met and exceeded in order to more accurately assess the potential bearing capacity of the NME stabilised material.

NME stabilisation allows for naturally available materials not conforming to the current traditional specifications to be effectively stabilised and utilised within a pavement structure in the upper pavement layers below the surfacing. The meeting of specifications with regard to the RCS and RTS protect the material particles from chemical weathering within the pavement structure. The former is a new concept introduced as an indication of resistance of the stabilised material to the effect of tyre pressures in association with water on an exposed layer. The specified RCS will delay the formation of deep potholes. This aspect has been observed in practice on pavement sections with damaged surfacings, with no increase in the depth of potholes over a rainy season for a period of 2 to 4 months (Figures 23 and 24).

Accelerated Pavement Testing (APT) results on a pavement structure with a damaged surfacing with free-flowing water over the exposed NME-treated base layer during testing have shown similar results [26]. A built-in resistance to pothole escalation (all too familiar on roads containing granular base layers protected by thin bituminous surfacings) is of high importance. Such behaviour trends will allow more time for maintenance teams to repair road surfacings before hazardous conditions develop, especially during rainy seasons and a lack of institutional capacity in terms of preventative periodic maintenance. These observations are to be confirmed and calibrated under laboratory conditions with comparisons between the RCS of various materials and traditional laboratory tests indicative of mechanical durability under wet-dry conditions.

The project specifications applicable to four categories of NME-stabilised naturally available materials (NME1 to NME4) are given in Figure 25. These recommendations take into consideration original APT tests performed on bitumen-emulsion-treated base layers [44], laboratory work performed on NME materials varying from relatively good to poor materials (including fine grained sands) [4,25], the experience obtained in practice through the use of NME-stabilised materials [24], the results of APT performed on two pavement structures [26,27] and the South African specifications for Bituminous Stabilised Materials (BSM) [43]. Adjustments are incorporated in the recommendations to allow for results obtained under ideal laboratory conditions and results to be obtained during

construction in the field for quality control (shown in red in Figure 25). As an additional safety factor, the RCS and RTS are also assessed in terms of the minimum requirements for the material category specified, further ensuring that the implementation of an NME road pavement structure is associated with a low-risk scenario, allowing for any practical challenges to be timely resolved.

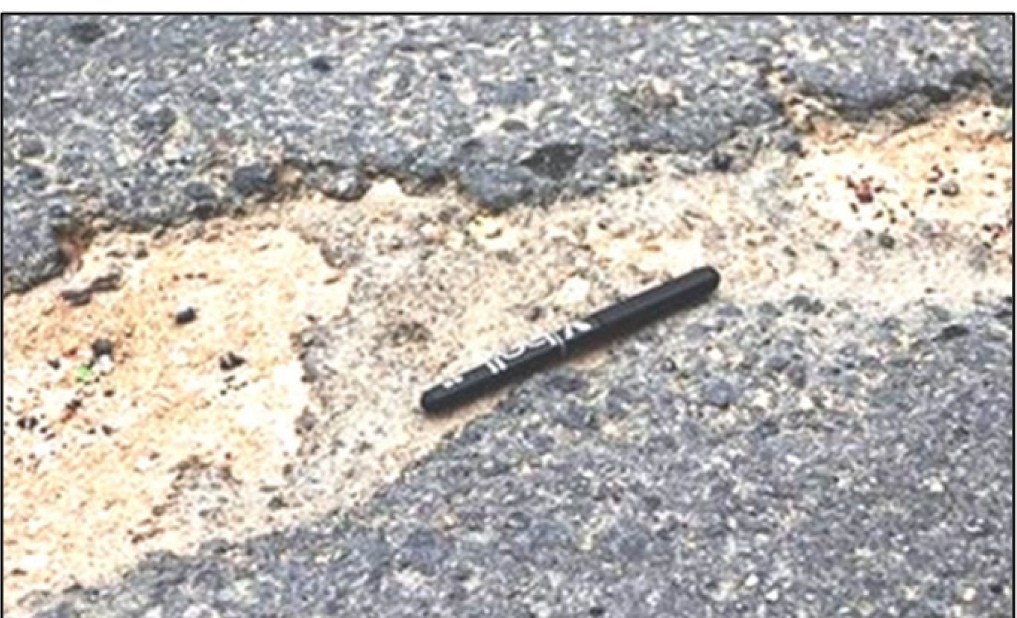

**Figure 23.** Stripping of asphalt (35 mm) (December 2019) due to placement of the asphalt in wet conditions (rain)—anionic NME stabilised base has been exposed for 2 months during the rainy season (Original material quality: CBR 15 to 25 @ 93% mod AASHTO) (Mpumalanga Province, South Africa).

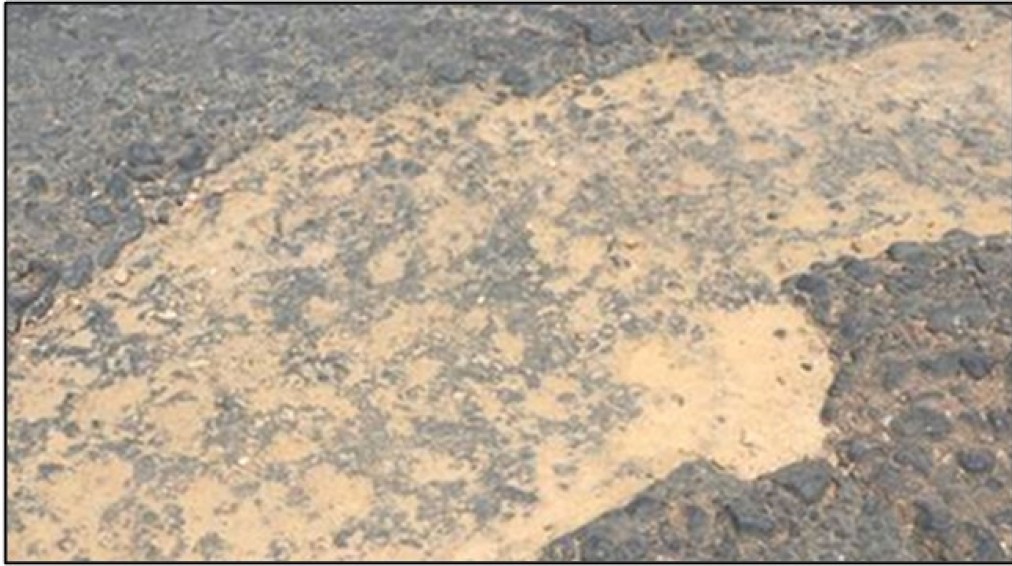

**Figure 24.** Stripping of surfacing (double seal with an elastomer modified binder) due to a contaminated binder (December 2018), resulting in little damage to an exposed NME stabilised base-layer after 2 months of traffic in the rainy season (in situ material originally of quality CBR > 10 @ 93% mod AASHTO) (Gauteng Province, South Africa).

| Test or Indicator | Material[1] | Material classification | | | |
|---|---|---|---|---|---|
| | | NME 1 | NME 2 | NME 3 | NME 4 |
| **Minimum material requirements before stabilisation and/or treatment (Natural materials)** | | | | | |
| Material spec.(minimum) Unestablished material: Soaked CBR[2] (%) (Mod AASHTO) | NG /(CS) | > 45[2] (95%) ACV < 30% | > 25[2] (95%) | > 10[2] (93%) | > 7[2] (93%) |
| Grading Modulus (GM) | NG | > 1.8 | > 1.5 | - | - |
| | GS | NA | > 1.5 | - | - |
| Sieve analysis: % < 0.075 mm ($P_{0.075}$) | ALL | < 20% | < 25 % | < 35 % | < 50 % |
| XRD scans: - Total sample - 0.075 mm fraction ($P_{0.075}$) | ALL ALL | √ √ | √ √ | √ √ | √ √ |
| % Material passing 2 µm ($P_{0.002}$) (e.g. Clay & Mica & Talc) as a % of Material (with Talc <10%) (XRD-scans of the material passing the 0.075 mm sieve is used to determine the % clay, mica (muscovite) and talc in the material – In this case $P_{0.002}$ = $P_{0.075}$ x ($P_{clay, etc.}$ in $P_{0.075}$.) | | NME stabilisation with micro-meter (µm) emulsion particle sizes | | | |
| | ALL | < 15 % | < 15 % | < 15 % | < 15 % |
| | | NME stabilisation with emulsion containing micro-scale as well as nano-scale particles (adjusted according to material grading) | | | |
| | ALL | NA | < 35 % | < 35 % | < 35 % |
| | | NME stabilisation with emulsion containing nano-scale and pico-scale particles (grading adjustments) together with technologies addressing workability of materials on site | | | |
| | ALL | NA | NA | > 35 % | > 35% |
| **Material specifications after stabilisation and/or treatment** | | | | | |
| In-situ density to be required after stabilisation and compaction (mod AASHTO) (%) (minimum) | Base | > 100 % | > 100 % | > 98 % | > 97 % |
| | Sub-base | NA | > 98 % | > 97 % | > 95 % |
| DCP(DN mm/blow)(Quality control) (stabilised and compacted) | | NA | NA | < 2.6 | < 3.5 |
| Mod AASHTO density (%) (for laboratory testing) | | > 100 % | > 100 % | > 100 % | > 100 % |
| *$UCS_{wet}$ (kPa) (150 mm Φ Sample) | **Design[3]** | **> 2 500** | **> 1 500** | **> 1 000** | **> 750** |
| | Construction[4] | > 2 200 | > 1 200[5] | > 700[5] | > 450[5] |
| Retained Compressive Strength (RCS): ($UCS_{wet}$/$UCS_{dry}$) (%) | | > 85 | > 75 | > 70 | > 65 |
| RCS in relation to minimum $UCS_{wet(criteria)}$ = $RCS_{effective}$ = (RCS x ($UCS_{wet}$/$UCS_{wet(criteria)}$)) (%) | | >100 | >100 | >100 | > 100 |
| *$ITS_{wet}$ (kPa) (150 mm Φ Sample) | **Design[3]** | **> 240** | **> 200** | **> 160** | **> 120** |
| | Construction[4] | > 220 | > 180[5] | > 140[5] | > 100[5] |
| Retained Tensile strength (RTS): $ITS_{wet}$/$ITS_{dry}$ (%) | | > 85 | > 75 | > 70 | > 65 |
| RTS in relation to minimum $ITS_{wet(criteria)}$ = $RTS_{effective}$ = ((RTS x ($ITS_{wet}$/$ITS_{wet(criteria)}$)) (%) | | >100 | >100 | >100 | > 100 |

[1]CS – crushed stone; NG – natural gravel; GS – gravel soil, and SSSC – sand, silty sand, silt, clay.\
[2]CBR only used as reference to traditionally used test procedures as a broad first indicator
*Definitions: UCS = Unconfined Compressive Strength; ITS = Indirect Tensile Strength);
  $UCS_{dry}$; $ITS_{dry}$ = testing after rapid curing; $UCS_{wet}$; $ITS_{wet}$ = testing after rapid curing and 4 hours in water (as per test procedure specified for the testing of cementitious stabilising agents (SANS 3001-GR32:2010, 2010));
  Design[3] = Minimum criteria to be met in the laboratory during the design phase
Construction[4] = Minimum criteria to be met during construction as part of quality control
[5]Criateria based on reference TG2 (Asphalt Academy, 2009)

**Figure 25.** Minimum recommended standard specifications for New-age (Nano) Modified (NME) stabilised materials, addressing four different classifications in terms of engineering requirements.

It should be noted that the limitations with regard to the minimum material properties that can effectively be used and treated as NME4 layers in pavement structures are yet to be established. Materials with a CBR as low as 7 at 93 per cent mod AASHTO [22,23] have been proven both in laboratories as well as in practice to be able to meet the minimum criteria for a NME4 layer with ease. Laboratory work is continuing with even poorer quality materials to establish the limits achievable with materials containing high percentages of clay, as demonstrated in Figure 25.

3.5.2. Recommended Test Protocols

In developing recommended test protocols, existing standards for the testing of traditional stabilising agents were strongly relied upon. In order to compare results, all materials need to be tested and evaluated using test protocols that are the same or taking into account certain limitations associated with specific stabilising agents. For example, the UCS$_{wet}$ and ITS$_{wet}$ of cementitious material are tested after soaking in water for 4 h—the same should apply for the comparison of any other stabilising agent. The whole approach is towards the development and recommendation of test protocols that are universally applicable for the evaluation of any NME stabilising agent and allow fair comparisons of results as required in an unbiased procurement process. In this regard, certain limitations need to be taken into account, for example:

- The time periods and temperature for a rapid curing process for the stabilisation should be standardised. However, it is well known that some polymers are damaged at temperatures exceeding 50 °C. Hence, the rapid curing process should be performed at temperatures between 40 °C and 45 °C (allowing for natural variations in remote or field laboratories as commonly experienced in practice);
- No special treatment of material samples, e.g., soaking in the stabilising agent after curing should be allowed;
- No treatment of the sample compaction moulds to assist with the removal of samples, e.g., the use of oil or grease, should be allowed, as this could influence the results obtained during the soaking of the sample in water;
- Rapid curing process pre- and post-treatments before testing must be standardised.

Considering the basic requirements, the following test protocols are currently recommended [13,18]. These recommendations are repeated due to the influence on test results to be discussed using the recommended design method:

1. The curing and testing process of the 152 mm diameter samples (127 mm high) shall be as follows: The prepared 152 mm diameter by a 127 mm height samples are to be prepared as per SANS PART 3001 GR50 [45] and GR51 [46] with some adjustments based on the requirements for a generalised procurement process, with no plastic covering. (Plastic covering is required when cement is included in the mix to assist in the hydration of the cement.) Samples are cured for 24 h in an oven at 30 °C before being subjected to a "rapid curing" process in an oven for 48 h at 40 °C to 45 °C (temperatures in the oven should NOT exceed 50 °C).
2. After 48 h, the samples must be removed from the oven and allowed to cool down for 24 h. This is to be preferably performed in the oven at 30 °C for 24 h.
3. Three (3) samples must be crushed to determine the ITS and UCS values. The values obtained are called the DRY ITS and the DRY UCS values.
4. Three (3) samples must be placed in a bath of water with a temperature of 22–25 °C for four (4) hours (as per test procedure specified for the testing of cementitious stabilising agents (SANS 3001-GR53 [47] and GR 54 [48], as adjusted)) and thereafter removed from the bath and allowed to drain off excess water before determining the WET ITS and WET UCS values.
5. If so approved by the Engineer, the "wet" tests (UCS and ITS) may suffice during the quality control during construction. For the lower-order roads (Category D and E), DCP tests performed at randomly selected spots may be approved for quality control as approved by the Engineer (refer to Figure 25).
6. During the design stage, three samples each must be preserved outside the moulds for a period of 28 days. After 28 days, the UCS (wet and dry), as well as the ITS (wet and dry), should be tested as per the procedure described above. The results of the 28-day tests should not show a decrease of more than 5 per cent in the values of the respective UCS and ITS tests as compared to that obtained after the rapid curing process.

7.     It is important to note that sample preparation must be performed in strict compliance with the prescribed procedures and NO deviation will be allowed, including:

     7.1     The moulds in which the samples are prepared are not to be treated with grease or any other lubricant to facilitate the easy removal of the sample, as this could influence the loss (during rapid curing) or increase (during soaking in water) of moisture and hence the measurements of UCS and ITS both in the dry and wet conditions;

     7.2     No additional soaking of samples in any "covering" liquid or any other material will be allowed, as this will make any comparison and application of test requirements invalid and not comparable to what is practically achievable during construction.

## 4. Typical Pavement Structural Designs Recommended for a Variation of Traffic Loadings and Traditionally Defined Subgrade Materials

Recommendations for typical pavement structures for different traffic loadings and subgrade conditions [13] were originally based on fatigue criteria developed for bitumen-emulsion-stabilised materials as developed from APT loading [44]. At the time, the designs were considered conservative but suitable for the introduction of NME-stabilising agents into a market known for its conservative approach. Subsequent APT loading performed on two roads [26,27] have confirmed that the designs are conservative, with ATP loading exceeding the design traffic loadings by some considerable margins. Consequently, a more optimum design catalogue is now recommended, shown in Figure 26 for different design traffic loadings (up to 30 million E80s) and subgrade conditions varying from in situ CBRs of 3 per cent at 93 per cent Mod AASHTO [22,23] and upwards. For comparisons with the traditional recommended designs, typical designs are compared in Figure 27. The work performed on the evaluation of thin chip seals, especially that performed on a Cape seal with a variation of modified binders, has confirmed the potential use of this thin surfacing as a protective surfacing layer for roads carrying relatively high traffic loadings on base and sub-base layers stabilised with a material compatible stabilising agent (with deformation characteristics comparable to that of asphalt surfacings specified for highways) [49].

Of particular importance is the fact that the behaviour of the NME stabilised material was found to be less sensitive to overloading than usually assumed for the assessment of pavement structures, with:

$$N = (P/80)^n \qquad (2)$$

where:

    N = Equivalent 80 kN dual wheel standard axle load;

    P = Applied dual wheel axle load;

    n = Damage coefficient (normally considered as 4.2 [50]).

In the case of the anionic NME stabilised layers, the damage coefficient has been found to be between 1.0 and 2.5. It follows that pavement layers stabilised with an anionic NME stabilising agent is not as sensitive to high wheel loads a per normal pavement structures. Hence, these pavements will be very suitable in an environment where law-enforcement in terms of over-loading is a scarcity or non-existent. Lifecycle cost analyses [51] on a number of projects have shown initial material cost savings of 30 to 50 per cent, with additional savings in construction time and considerable savings in possible periodic maintenance over the design period [25,26,49].

For comparative purposes, the recommended designs based on NME stabilised naturally available materials are compared with traditional designs using high-quality crushed stone layers and/or Cement (C)-treated layers [52], as shown in Figure 27. Material classifications are as per the South African Technical Recommendations for Highways document TRH14 [53].

| Design traffic Loading (max) Million Equivalent 80kN Standard Axles ($10^6$ E80s) | Typical road Category | RECOMMENDED Pavement structure with naturally available materials stabilised with anionic New-age (Nano) Modified Emulsion (NME) (meeting the minunum specifications for the stabilised material class, i.e. NME1 to NME4) | | | |
|---|---|---|---|---|---|
| $30 \times 10^6$ E80s | A | Surfacing | 20 mm Cape Seal[1] / 30 mm Asphalt (modified binder)[2] | | |
| | | Base | 150 mm NME1 | 150 mm NME1 | 150 mm NME1 |
| | | Sub-base | 100 mm NME3 | 150 mmm NME3 | 150 mm NME3 |
| | | Upper Selected | 150 mm NG (Minimum CBR = 25 @ 95% mod. AASHTO) | 150 mm NG (Minimum CBR = 15 @ 93% mod. AASHTO) | 200 mm NG (Minimum CBR = 15 @ 93% mod.AASHTO) |
| | | Sub-grade | Gravel/Soil (Minimum CBR = 7 @ 93% mod. AASHTO) | | Gravel/Soil (Minimum CBR = 3% @ 93% mod. AASHTO) |
| $10 \times 10^6$ E80s | A/B | Surfacing | 20 mm Cape Seal[1] / 30 mm Asphalt (modified binder)[2] | | |
| | | Base | 100 mm NME2 | 100 mm NME2 | 150 mm NME2 |
| | | Su-bbase | 100 mm NME4 | 150 mm NME4 | 150 mm NME4 |
| | | Upper Selected | 150 mm NG (Minimum CBR = 25 @ 95% mod. AASHTO) | 150 mm NG (Minimum CBR = 15 @ 93% mod. AASHTO) | Gravel/Soil (Minimum CBR = 3% @ 93% mod. AASHTO) |
| | | Sub-grade | Gravel/Soil (Minimum CBR = 7 @ 93% mod. AASHTO) | | |
| $3.0 \times 10^6$ E80s | B | Surfacing* | Cape Seal (10/14/20 CS[1]) / Double Seal / 25 mm Asphalt (modified binder)[2] | | |
| | | Base | 100 mm NME4 | 150 mm NME 4 | 200 NME4 |
| | | Sub-base | 150 mm NG (Minimum CBR = 25 @ 95% mod. AASHTO) | 150 mm NG (Minimum CBR = 15 @ 93% mod. AASHTO) | Gravel/Soil (Minimum CBR = 3% @ 93% mod. AASHTO |
| | | Subgrade | Gravel/Soil (Minimum CBR = 7 @ 93% mod. AASHTO) | | |
| $< 1.0 \times 10^6$ E80s | C | Surfacing* | Cape Seal (10/14/20 CS[1]) / Double Seal / 30 mm Asphalt (modified binder)[2] | | |
| | | Base | 100 mm NME4 | 100 mm NME4 | 150 mm NME4 |
| | | Sub-base | 150 mm NG (Minimum CBR = 25 @ 95% mod. AASHTO) | 150 mm NG (Minimum CBR = 15 @ 93% mod. AASHTO) | Gravel/Soil (Minimum CBR = 3% @ 93% mod. AASHTO) |
| | | Sub-grade | Gravel/Soil (Minimum CBR = 7 @ 93% mod. AASHTO) | | |

\* Appropriate Seal according to requirements(Urban/Rural/Required surfacing life/Labour intensive constuction/etc);

[1] Recommended Seal: Cape Seal with a 10 mm or 14 mm stone or 20 mm stone with NME slurry and no cement as a filler;

[2] Binder with best proven modification to protect against oxidation ("aging"), deformation and cracking for the specific climatic zone - asphalt can be considered for an equavalent reduction in the thickness of the NME layer.

**Figure 26.** Typical recommended pavement structures developed for the construction of road pavements using a material compatible anionic New-age (Nano) Modified Emulsion (NME) stabilising agent with naturally available materials as per specifications (Figure 25).

| Loading+B2:O51 | | Recommended Pavement structure (Material classification - draft TRH14 (1987) - Layer thicknesses in mm) | | | | Alternative BSM or NME design Pavement structure with BSM design (Bitumen Emulsion) using appropriate materials or alternatively naturally available (in-situ) materials stabilised with Modified Emulsions[2] meeting the minunum specifications for the stabilised material class | | |
| Million Equivalent Standard Axles | Typical road Categor... | (draft TRH4)* Granular Base (Dry) | (draft TRH4)* Granular Base (Wet) | (draft TRH4)* Hot-Mix Asphalt | (draft TRH4)* Cemented Base | | | |
| ES30 | A | 40 A / 150 G1 / 125 C3 / 125 C3 / G7 | 50 A / 150 G1 / 200 C3 / 200 C3 / G7 | 40 A / 120 BC / 200 C3 / 200C3 / G7 | | 20 mm Cape Seal (CS[1])/ Double seal / 30 mm Aspahalt[2] 150 NME1 / 100 NME3 / 150 G6 / G9 | 150 NME1 / 150 NME3 / 150 G7 / G9 | 150 NME1 / 150 NME3 / 200 G7 / G10 |
| ES10 | A/B | 40 A / 150 G2 / 125 C3 / 125 C3 / G7 | 40 A / 150 G1 / 150 C3 / 150 C3 / G7 | 40 A / 90 BC / 150 C3 / 150 C3 / G7 | | 20 mm Cape Seal (CS[1])/ Double seal / 30 mm Aspahalt[2] 100 NME2 / 100 NME4 / 150 G6 / G9 | 100 NME2 / 150 NME4 / 150 G7 / G9 | 150 NME2 / 150 NME4 / G10 |
| ES3.0 | B | S*/30 A / 150 G3 / 150 C4 / G7 | S/30 A / 150 G1 / 150 C3 / G7 | 30 A / 80 BC / 200 C4 / G7 | S (40 A) / 125 C3 / 200 C4 / G7 | ** Cape Seal /(10/14/20 CS[1]) / Double seal / 25 mm Aspahalt[2] 100 NME4 / 150 G6 / G9 | 150 NME4 / 150 G7 / G9 | 200 NME4 / G10 |
| < ES1.0 | C | S (40 A) / 125 G4 / 125 C4 / G7 | S (40 A) / 125 G2 / 150 C4 / G7 | | S (40 A) / 125 C3 / 125 C4 / G7 | ** Cape Seal(10/14/20 CS[1]) /Double seal / 25 mm Asphalt[2] 100 NME4 / 100 G6 / G9 | 100 NME4 / 150 G7 / G9 | 150 NME4 / G10 |

*Typical examples of recommended pavement structures taken from the draft TRH4 [52] with material classification according to TRH14 [53]

** Appropriate Seal according to requirements(Urban/Rural/Required surfacing life/Labour intensive constuction/etc);

[1] Recommended Seal: Cape Seal with a 10 mm or 14 mm stone or 20 mm stone;

[2]Binder with best proven modification to protect against oxidation ("aging"), deformation and cracking for the specific climatic zone - asphalt in the place of a seal can be considered for an equavalent reduction in the thickness of the NME layer.

**Figure 27.** Comparative pavement designs based on high-quality crushed stone and/or cement-treated materials with relatively thin surfacings and the newly recommended NME stabilised designs using naturally available materials.

## 5. Practical Application and Comparison of Material Design Methods/Approaches

### 5.1. Design Method: Comparison of Results

The design approach, as detailed, was recently (2019) used on a road in South Africa to complete an alternative design using locally available materials not conforming to the traditionally required standards. The original pavement design consisted of a 150 mm cement-stabilised sub-base with a UCS of between 1500 MPa and 3000 MPa and a 150-mm high-quality crushed stone base (G1 [53]) compacted to a minimum of 102% mod AASHTO [22,23] (88 per cent ARD) with a 40-mm asphalt surfacing. No high-quality stone is available in this part of the sub-continent, and an alternative design was proposed using NME stabilised naturally available materials.

After discussions with the client, an alternative design was approved, specifying a 200-mm NME3 quality base layer (refer Figure 25) with a 40-mm asphalt surfacing (not recommended, but preferred by the client) with a modified binder. The naturally available material has a shortage in fines with little cohesion and a CBR of between 15 and 25 at 93 per cent mod-AASHTO [22,23]. Suppliers were invited to submit stabilising agents for testing and pricing using the naturally available materials, meeting the minimum requirements for an NME3 quality pavement layer. Three suppliers responded, and their products were tested and evaluated by an independent laboratory against the engineering requirements contained in Figure 25. The three suppliers used different design approaches and provided different products to stabilise the materials. The test procedures previously referred to as standard test methods (refer Section 3.5.2) were followed to evaluate all the different products. The results and evaluation of the test results are given in Figure 28 (comparison of the Bituminous Stabilised Materials (BSM) [54], Trial-and-Error (T & E) approach and Mineralogy Chemistry NME (as detailed in this article) design methods).

The results highlighted in green show test results exceeding the minimum specifications. Similarly, the results highlighted in red indicate that the minimum engineering requirements are not being met. The results shown in blue are the test results (dry conditions after rapid curing) required to determine the RCS and the RTS as defined in Figure 25. It is shown that only two of the suppliers could meet all the engineering requirements. The material costs of the design approach based on the Mineralogy of the materials, as detailed in this article, gave results exceeding the required engineering minimum criteria by some considerable margin. In addition, the comparative cost was less than 50 per cent of the only other stabilising agent that met all the engineering requirements.

### 5.2. Discussion of Results

The considerable advantage of the materials design approach based on the basic scientific approach as contained in this article, using materials of relatively poor quality, is irrefutable. Of some interest is the Retained Compressive Strength (RCT) and Retained Tensile Strength (RTS) of more than 100 per cent, shown in Figure 28, using the design approach based on the scientific approach of mineralogy of the materials (last two lines of the results shown in Figure 28). These results are unheard of in pavement engineering, with an immediate reaction of erroneous and inaccurate laboratory work and/or recording from practitioners. However, these results are not unusual and are easily explained and understood if the basis of the NME stabilisation process, the basis of the modification, the material characteristics and the test procedure as contained in Section 3.5.2 are well known.

With a scientifically based material-compatible designed NME stabilising agent, it is not unusual to achieve a 100 per cent hydrophobicity (or very close to 100 per cent) of your test sample. The effect is that the submersions of the test sample after completion of the rapid curing process in a water bath will result in basically zero penetration of water into the sample, not negatively effecting the engineering properties to be tested.

PROJECT: UPGRADING OF GRAVEL ROAD: South Africa

TEST RESULTS: UPGRADING OF GRAVEL ROAD USING IN-SITU MATERIAL CBR @ 93 % mod AASHTO between 15 and 45

MATERIAL REQUIREMENT: NME3

REQUIRED CRITERIA:

| | | | |
|---|---|---|---|
| $UCS_{wet} > $ (kPa) | 1000 | Retained Compressive Strength (RCS) (% ($UCS_{wet}/UCS_{dry}$)) > | 70 |
| | | RCS in relation to minimum criteria in a wet condition (RCS-MC) (RCS x ($UCS_{wet}/UCS_{wet-criteria}$) (%)) > | 100 |
| $ITS_{wet} > $ (kPa) | 160 | Retained Tensile Strengthcohesion (RTS) (% ($ITS_{wet}/ITS_{dry}$) > | 70 |
| | | RTS in relation to minimum criteria in a wet condition (RTS x ($ITS_{wet}/ITS_{wet-criteria}$)) > | 100 |

| No | Product tested | Design Methodology | Supplier | $UCS_{dry}$ kPa | $UCS_{wet}$ kPa | RCS % | RCS-MC % | $ITS_{dry}$ kPa | $ITS_{wet}$ kPa | RTS % | RTS-MC % | Comments | Cost/m3 (Product)* |
|---|---|---|---|---|---|---|---|---|---|---|---|---|---|
| 1 | 2% Roadcem + 2.5% SS60 (55lts/m3) (BSM) | BSM¹ | 1 | 2351 | 1790 | 76 | 136 | 289 | 217 | 75 | 102 | Pass | US$ 49.40 |
| 2 | 2% Roadcem + 1.5 % SS60 (33lts/m3) (BSM) | BSM¹ | 1 | 1585 | 1166 | 74 | 86 | 171 | 105 | 61 | 40 | Fail | |
| 3 | 2% Roadcem + 1.0 % SS60 (22lts/m3) (BSM) | BSM¹ | 1 | 1409 | 887 | 63 | 56 | 121 | 48 | 40 | 12 | Fail | |
| 4 | 1.0 % Cationic NME (22 lts/m3) | T & E² | 1 | 1238 | 705 | 57 | 40 | 108 | 29 | 27 | 5 | Fail | |
| 5 | 1.5 % Cationic NME (33 lts/m3) | T & E² | 1 | 1308 | 739 | 56 | 42 | 112 | 27 | 24 | 4 | Fail | |
| 6 | 2.5 % Cationic NME (55 lts/m3) | T & E² | 1 | 1370 | 798 | 58 | 46 | 138 | 37 | 27 | 6 | Fail | |
| 7 | 2% Roadcem + 2.5 % SS60 (55lts/m3) (BSM) | BSM¹ | 2 | 2086 | 1515 | 73 | 110 | 233 | 166 | 71 | 74 | Fail | US$ 52.50 |
| 8 | 2% Roadcem + 1.5 % SS60 (33lts/m3) (BSM) | BSM¹ | 2 | 1565 | 971 | 62 | 60 | 139 | 86 | 62 | 33 | Fail | |
| 8 | 2% Roadcem + 1.0 % SS60 (22lts/m3) (BSM) | BSM¹ | 2 | 1396 | 823 | 59 | 49 | 132 | 45 | 34 | 10 | Fail | |
| 10 | 1.0 % Cationic NME (22 lts/m3) | T & E² | 2 | 1253 | 707 | 56 | 40 | 105 | 23 | 22 | 3 | Fail | |
| 11 | 1.5 % Cationic NME (33lts/m3) | T & E² | 2 | 1329 | 746 | 56 | 42 | 135 | 29 | 21 | 4 | Fail | |
| 12 | 2.5 % Cationic NME (55lts/m3) | T & E² | 2 | 1396 | 769 | 55 | 42 | 162 | 39 | 24 | 6 | Fail | |
| 13 | 1.0 % Anionic NME (22lts/m3)(Double Emulsification) | Mineralogy³ | 3 | 3110 | 3120 | 100 | 313 | 281 | 356 | 127 | 282 | Pass | US$ 24.00 |
| 14 | 1.0 % Anionic NME (22 lts/m3)(Single Emulsification) | Mineralogy³ | 3 | 2090 | 2840 | 136 | 386 | 181 | 283 | 156 | 277 | Pass | US$ 23.40 |

BSM¹ = Bituminous Stabilased Material (BSM) design approach - Industry Manuel - TG2 [54]

T & E² = Trail and Error design adding different percentages of stabilising agents as per supplier

Mineralogy³ = Design based on mineralogy of materials and chemical interaction with Nano-Modified Emulsion (NME) Stabilising agent

*Price provided by contractor on site South African Rand conversion: 1 US$ = R16 South African Rand

**Figure 28.** Results of the testing of stabilised materials as per design criteria for an NME3 (Figure 24) layer using different stabilising agents and different design methods/approaches, as detailed at the bottom.

However, during the last phase of the rapid curing process, the test samples are left in the oven for 24 h at 30 °C to allow the samples to "cool down" (as per current test recommendations). The modified NME stabilising agent has as a basic ingredient of bitumen (or equivalent polymer) that is added to the material in the form of a modified emulsion. During the rapid curing process, the bitumen (or equivalent polymer) in the emulsion separates from the water and the water is effectively repelled from the test sample by the nano-silane modification that attaches the bitumen particle "permanently" to the material particles in the material test sample and effectively coats every material particle to be hydrophobic.

It is a well-known fact that the effective modulus ("stiffness" or viscosity) of bitumen (or equivalent polymer) is a function of temperature. The water in the water bath should be between 22 °C and 25 °C (in practice, the temperature of the water is often lower in the order of 20 °C to 22 °C). Hence, during the submersions of the test samples in the water-bath, the samples are taken from an environment of 30 °C (from the oven) and submersed into water at a temperature of 22 °C to 25 °C (or even lower). The effective modulus (stiffness or viscosity) of the bitumen (or equivalent polymer) at a lower temperature will increase significantly. With water being unable to penetrate the sample due to the hydrophobicity, the higher effective modulus (stiffness of the binder) will result in higher test results after submersion in the water bath. Hence, under these conditions, using a scientifically designed material-compatible NME stabilising agent, the wet test results will often (as has been verified on numerous occasions) be higher than the dry results. This is only achieved when the possible negative impact of water is neutralised by the hydrophobicity introduced into the material using an effective nano-silane modification (material compatible design).

In order to obtain a true indication of the RCS and RTS (compare dry and wet test results), the final curing (24 h) and the temperature of the water should be closely matched. Hence, it is be recommended that the test procedure be changed and that the final 24 h in be oven should be performed at 20 °C and not the currently recommended 30 °C.

## 6. Conclusions

The use of nanotechnology solutions in the built environment is nothing new. Scientists in Europe have been tasked with developing products to protect stone buildings against the effect of water and pollution since the early 1800s. Initially, these scientists were faced with contrasting results and success. The variation in the results, applying various silane-based (nano-size) treatments that were developed at the time, soon led the scientists to conclude that the type of stone and the condition of the stone are crucial information to take into consideration when using and applying any specific product. The products developed and the lessons learnt over more than 150 years in the built environment for the protection of buildings, monuments and statues are also directly applicable to the field of road pavement materials design.

This article details a material design approach incorporating basic science in terms of the chemical interaction between the mineralogy of naturally available materials and available, proven nanotechnology solutions, enabling the use of traditionally classified "marginal" materials in the upper layers of a road pavement structures below the surfacing, at a low risk. Naturally available materials normally do not conform to the criteria that are traditionally applied for the evaluation of materials considered suitable for use in these layers. These traditional material classifications are, to a large extent, based on empirically derived tests far removed from the basic petrological genetics of materials. These inconsistencies have been recognised and pointed out by geologists since the 1950s, with little impact on traditional approaches used by pavement engineers for the classification of road building materials.

The successful implementation of new-age nanotechnologies in road pavement materials design will depend on the scientific analysis of the primary minerals (type of stone) as well as the secondary minerals (condition of the stone) comprising the naturally available

materials. In order to minimise risk, a materials design approach enabling the use of naturally available materials is presented, based on the mineralogy of the materials and the chemical interaction thereof with any modified stabilising agent. The scientifically based material design method/approach ensures that the NME stabilising agent is material compatible and that it will be able to develop strong chemical bonds with the material particles as well as the stabilising agent. The implementation of a scientifically based materials design approach is universally applicable, in contrast to empirically derived methods that require verification under different conditions.

The materials design method bases the selection of a material compatible NME stabilising agent on the primary as well as the secondary minerals. Adjustments in the various volumes and use of various applicable new-age nanotechnologies are performed using a few easily obtainable material parameters. The aim of the nano-silane modification is also to neutralise any possible negative impact of the presence of secondary minerals in the naturally available granular materials on the future behaviour of the pavement structure. The secondary minerals are neutralised by ensuring that the material particles are chemically changed to become hydrophobic, not allowing water access to these minerals. The chemical decomposition of materials is only possible in the presence of water. If water is largely repelled from the material particles, further chemical weathering is limited and durability is improved.

The NME stabilised material's properties are assessed against engineering criteria indicative of the physics of the material behaviour, i.e., the stresses and strains that need to be tolerated by the stabilised materials, as well as the future required durability, i.e., the level to which these properties will be maintained within the NME stabilised material over time. These criteria are specified using a classification system for the NME stabilised materials as a function of the required bearing capacity within the pavement structure in terms of the required engineering stresses and strains, in line with modern Mechanistic-Empirical pavement design methods.

This approach requires pavement engineers to have a basic understanding of mineralogy, chemical weathering and the influence thereof on the materials and the selection of a material compatible NME stabilising agent. These aspects are addressed within this article. Chemical weathering is directly associated with the higher temperature and high rainfall areas of the world, especially concentrated in, but not exclusive to, areas within the Tropics, on either side of the Equator. These regions are also associated with the majority of the developing world, which is urgently in need of improved transportation infrastructure in support of its developing economies.

The practical implementation of the presented design approach has shown material cost savings of between 30 and 50 per cent with similar savings in construction times when compared to traditionally used pavement designs. The alternative designs using naturally available materials stabilised with material compatible NME stabilising agents, have been evaluated using numerous laboratory tests as input into the recommended material classification system of NME stabilised materials. An updated catalogue of designs, initially developed using Mechanistic-Empirical design methods, were refined using these laboratory tests in combination with full-scale APT results, enabling the designs to be optimised and a revised design catalogue to be compiled.

The implementation of the material design approach has recently been evaluated in practice against traditional design approaches. The use of the mineralogy-based NME stabilisation of the available materials not only far exceeded the required engineering criteria, but also realised a comparative saving in material costs of more than 50 per cent. Similar savings in the construction time can be realised. Additional requirements addressed in project specifications to further reduce possible associated risks during construction will make the scientifically-based materials design method/approach as detailed an attractive alternative.

**Author Contributions:** G.J.J. under the directive of the Head of Department of Civil Engineering, W.J.v.S., has been leading the research into the provision of affordable road infrastructure at the faculty of Engineering, University of Pretoria. W.J.v.S. recognized the potential of nanotechnology solution in the field of pavement engineering more than a decade ago. G.J.J., through involvement in the private sector and the support of road authorities, has been instrumental in the development of scientific principles, ensuring that implementation can be achieved at a minimum risk. All authors have read and agreed to the published version of the manuscript.

**Funding:** This research received no external funding.

**Institutional Review Board Statement:** Not applicable.

**Informed Consent Statement:** Not applicable.

**Data Availability Statement:** Not applicable.

**Acknowledgments:** The support of GeoNANO Technologies (Pty) Ltd., 18 Davies road, Wychwood, Germiston 1401, South Africa, Tel.: +27-844078489, www.geonano.co.za (accessed on 13 August 2021), info@geonano.co.za, in support of students in the Department of Civil Engineering, University of Pretoria, Pretoria, South Africa, to test a wide variety of materials as part of final year projects and post-graduate theses, testing the various principles identified in this paper, is acknowledged.

**Conflicts of Interest:** The authors declare no conflict of interest.

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
