# Peer review of "Nanotechnology Incorporation into Road Pavement Design Based on Scientific Principles of Materials Chemistry and Engineering Physics Using New-Age (Nano) Modified Emulsion (NME) Stabilisation/Enhancement of Granular Materials"

_applsci, doi:10.3390/app11188525_

Round 1
Reviewer 1 Report
It is good work. The paper needs thorough editing to improve the technical writing. Also, there are a few spelling errors in the paper.
The paper balance is too much on the material description while it will be helpful for the reader to provide some examples of using such materials and improvements in the design in various situations.
Also, the material characterization part is lacking in the current version. What moduli values can be achieved after treating the geomaterials? It will be useful to compare the stiffness values with and without treatment for the same materials.
How the drainage properties (permeability) are affected by treatments should be also included in the results.
What are the impacts of these treated materials on the pavement system? Some examples will be helpful for the readers to access the life cycle costs.
Reviewer 2 Report
In this manuscript, the author attempts to develop a basic materials design method to ensure that nano-silane modified emulsions can be successfully used in the construction of pavement layer using naturally available materials with low risk. This method take identification of scientifically material characteristics, identification of a material compatible NME stabilising agent, and required fundamental engineering properties into consideration, and some examples are given to explain how the method works. However, before submitting for publication, there are still some questions needed to be clarified, as follows.
(1)It is mentioned in lines 493 to 496 that for material with high percentage of clay, the stabilizing agent should have graded to match the increase in the area to be covered. However, Figure 12 cannot clearly express this, and the content that the article wants to reflect cannot be expressed intuitively. Please consider changing the form of Figure 12 to highlight the difference among material with high percentage of clay and ordinary materials and material with fine grading.
(2)In the example given in Figure 18, the description from line 634 to line 636 indicates: the effective % passing through the 0.075 mm sieve will be multiplied by a factor of 2.5 to an effective 40 percent. However, this content does not match the information in the figure, please consider making corrections.
(3)Many figures in the article are cited incorrectly. For example, the source of Figure 2 on line 318 is incorrect, Figure 15 cited in line 638, Figure 14 and Figure 15 cited in line 673 are not consistent with the content expressed, please carefully check and modify. In addition, figure 6,figure 19, figure 21, figure 24 and many other figures have low definition, please consider modifying.
(4)It is mentioned in lines 588 to 590 that CBR is not of importance with the selection of an applicable nanotechnology solution, it is just as a reference to traditionally used test methods. Please explain the reasons for choosing CBR value as a reference for traditional test methods.
(5)The conclusion part of the article is too long and contains a lot of repetitive content. Please consider simplifying it appropriately.
Reviewer 3 Report
The article “Nanotechnology Incorporation into Road Pavement Design based on Scientific Principles of Materials Chemistry and Engineering Physics” deals with a very interesting topic. The manuscript it is also well written. However, I suggest addressing the following aspects for publication:
- Precise the article title. The manuscript deals with the methodology to evaluate the use of Nano Modified Emulsions (NME) in layers of aggregates forming flexible pavements. So, as it is now “Nanotechnology Incorporation into Road Pavement Design based on Scientific Principles of Materials Chemistry and Engineering Physics”, it is to general. Nanomaterials include a broader spectrum (e.g. nanoparticles, nanofibers, nanocoatings) which has different functional properties of pavements.
- Using references, confirm this sentence “Most work throughout the world on the use of nanotechnology solutions in pavement engineering has been concentrated on the improvement of asphaltic materials”. Also, there is a lot of evidence of nanomaterials in rigid pavements.
- Improve the quality of images. Also, verify copyright issues.
- Improve conclusions. They are too general. As reader, I expected to find the differences with the proposed method and the traditional ones.
